# The Synergy of LLMs & RL Unlocks Offline Learning of Generalizable Language-Conditioned Policies with Low-fidelity Data

**Thomas Pouplin** [* 1]  **Katarzyna Kobalczyk** [* 1]  **Hao Sun** [1]  **Mihaela van der Schaar** [1]

## Abstract

Developing autonomous agents capable of performing complex, multi-step decision-making tasks specified in natural language remains a significant challenge, particularly in realistic settings where labeled data is scarce and real-time experimentation is impractical. Existing reinforcement learning (RL) approaches often struggle to generalize to unseen goals and states, limiting their applicability. In this paper, we introduce *TEDUO*, a novel training pipeline for offline language-conditioned policy learning in symbolic environments. Unlike conventional methods, *TEDUO* operates on readily available, unlabeled datasets and addresses the challenge of generalization to previously unseen goals and states. Our approach harnesses large language models (LLMs) in a dual capacity: first, as automatization tools augmenting offline datasets with richer annotations, and second, as generalizable instruction-following agents. Empirical results demonstrate that *TEDUO* achieves data-efficient learning of robust language-conditioned policies, accomplishing tasks beyond the reach of conventional RL frameworks or out-of-the-box LLMs alone.

## 1. Introduction

**Motivation.** Enabling AI agents to follow human-provided natural language instructions is a critical step toward building intelligent systems that can perform complex, multi-step tasks in dynamic environments. However, existing methods often require vast quantities of annotated data or rely on costly online interactions with the environment to train effective policies. Such requirements limit their scalability to real-world scenarios where collecting annotated or interac-

tion data is infeasible. Moreover, current RL-based agents often struggle to generalize to new goals and environments beyond their training data (Yang et al., 2023). These challenges underline the need for a change of focus in the area of instruction-following agents: from reliance on online interactions in environments with rich forms of feedback to the learning of generalizable policies from unlabeled observational data. We illustrate the problem setting considered with the following hypothetical scenario:

> Imagine recording a human using their smartphone in daily tasks, such as text messaging or calendar management. How can we train an an AI assistant following natural language instructions, e.g.*"Book a restaurant for 7 PM"* or *"Send an email to John."*, by just recording the actions taken by the human and not having to manually label each state or action recorded? How can we make the learned policies generalize to commands that have not been explicitly observed in the data?

**Problem setting.** We formalize our setup by framing it as an offline language-conditioned policy learning problem. We assume access to a pre-collected dataset of state-action transitions, $\mathcal{D}$ consisting of triplets $(x, a, x')$, where $x$ and $x'$ belong to an observable state space $\mathcal{X}$, and $a$ represents actions within an action space $\mathcal{A}$. To enable the grounding of the environment dynamics with the space of natural language we additionally require: 1) that the individual states and actions are representable in a textual format; 2) access to an unordered list of goals, $\mathcal{G}$ expressed in natural language that are plausibly achievable within the environment of interest. We posit that both $\mathcal{D}$ and $\mathcal{G}$ are often easy to obtain by simply recording agents interact with the environment and curating a list of natural language commands corresponding to the tasks typically performed within that environment. Without constraining assumptions regarding the optimality of the data collection policy with respect to the goals $\mathcal{G}$, nor access to the ground-truth state-transition dynamics or rewards, our aim is to learn a language-conditioned policy, $\pi^*$, that can determine optimal actions with respect to new states $x \notin \mathcal{D}$ and new goals $g^* \notin \mathcal{G}^{tr}$, where $\mathcal{G}^{tr} \subset \mathcal{G}$ is the subset of goals used for training (see section 2 for details).

**Challenges.** The task of learning $\pi^*$ from $\mathcal{D}$ and $\mathcal{G}^{tr}$ alone

---

[*]Equal contribution  [1]Department of Applied Mathematics and Theoretical Physics, University of Cambridge, United Kingdom. Correspondence to: Thomas Pouplin <tp531@cam.ac.uk>, Katarzyna Kobalczyk <knk25@cam.ac.uk>.

*Proceedings of the 42$^{nd}$ International Conference on Machine Learning*, Vancouver, Canada. PMLR 267, 2025. Copyright 2025 by the author(s).

might seem impossible without resorting to human supervision. We can immediately identify the following challenges: **C1) Unlabeled data.** The dataset $\mathcal{D}$ lacks explicit labels linking states $x \in \mathcal{X}$ to the goals $g \in \mathcal{G}$. Nor does it include any rewards indicating the optimality of actions in relation to these goals. **C2) Limited exploration.** We are in an offline setup with our knowledge of the environment dynamics being constrained to the state-action transitions observed in $\mathcal{D}$. **C3) Unknown data collection.** We make no assumptions regarding the optimality of the data collection policy with respect to the training or testing goals. The actions in $\mathcal{D}$ could be entirely random or generated by policies aimed at solving goals with an unknown relationship to those in $\mathcal{G}$. **C4) Generalization.** Beyond solving goals from $\mathcal{G}^{tr}$ and taking optimal actions in previously observed states $x \in \mathcal{D}$, we want our agent to generalize to new states and language commands corresponding to genuinely novel goal states.

**Solution: LLMs & RL synergy.** Recent advances in LLMs offer a promising solution to these challenges. LLMs, pre-trained on vast amounts of Internet data, possess the requisite prior knowledge to understand natural language and follow simple instructions. However, while LLMs excel at general language comprehension, their ungrounded knowledge is insufficient for executing complex, multi-step decision-making tasks in dynamic environments (Finn, 2024; Szot et al., 2024). In this paper, we propose a novel, sequential training pipeline for offline language-conditioned policy learning—*TEDUO: Teaching the Environment Dynamics from Unlabeled Observations*. As displayed in Figure 1, TEDUO distills knowledge of the environment dynamics into a pre-trained LLM through supervised fine-tuning (SFT). This knowledge is obtained by learning optimal policies with traditional RL, based on the offline dataset augmented with LLM-generated labels and optional state abstractions. Within TEDUO, LLMs fulfill the dual role of *cheap data enhancers* and *flexible generalizers*, elevating conventional RL to address challenges C1-C4.

**Contributions.** We categorize our contributions as having a significant impact both for the fields of LLMs and RL.

*Significance for the LLM Community:* ▶ **Grounding LLMs for Multi-Step Decision Making:** In line with previous research, we demonstrate that standalone LLMs surprisingly fail even in simple multi-step decision-making tasks. We identify this limitation as stemming from the lack of grounding in environment dynamics. Crucially, we show that such grounding can be successfully achieved through SFT, enabling LLMs to integrate their background knowledge with actionable policies. ▶ **Core Skill Acquisition:** Our analysis reveals that fine-tuned LLMs acquire core decision-making skills rather than merely memorizing optimal actions. This highlights the potential of LLMs to generalize effectively to previously unsolved tasks.

*Significance for the RL Community:* ▶ **Offline Learning from Low-fidelity Data**: We introduce the first method to enable learning generalizable language-conditioned policies using only an unlabeled dataset of state-action transitions and an unpaired set of language commands. This addresses a critical bottleneck of conventional RL by eliminating the need for expensive labeled datasets or real-time experimentation. ▶ **Enhanced Generalization and Data Efficiency:** TEDUO significantly improves both the generalization capacity and data efficiency of offline training, outperforming conventional RL approaches. We provide empirical evidence that our method scales effectively and facilitates robust policy learning in symbolic environments.

## 2. Problem Formalism

We are given a dataset $\mathcal{D}$ of past interactions of an agent acting according to a data collection policy $\pi^\beta$. This dataset is represented as a collection of trajectories:

$$\mathcal{D} = \{\tau_i\}_{i \in \mathcal{I}}, \ \tau_i = \{(x_t, a_t, x_{t+1})\}_{t=0}^{T_i},$$
$$x_0 \sim \rho, \ x_{t+1} \sim P(\cdot|x_t, a_t), \ a_t \sim \pi^\beta(\cdot|x_t),$$

where $P$ is the state transition function determining the next state given an action $a_t \in \mathcal{A}$ and state $x_t \in \mathcal{X}$ and $\rho$ represents the distribution of initial states. Alongside $\mathcal{D}$, we are given an unpaired set of goals $\mathcal{G}$ split into training $\mathcal{G}^{tr}$ and testing $\mathcal{G}^{test}$ subsets, describing in natural language tasks an agent may attempt to solve within the environment.

Denoting by $\mathcal{P}(\mathcal{X})$ the powerset of $\mathcal{X}$, we assume there exists a ground-truth mapping $\phi : \mathcal{G} \to \mathcal{P}(\mathcal{X})$ associating each goal $g$ with a subset of the state space, $\phi(g) = \mathcal{X}_g \subseteq \mathcal{X}$. We say that $g$ is achieved at time step $t$, if $x_t$ lies in $\mathcal{X}_g$.[1] Then, the cumulative discounted reward: $\sum_{t=0}^\infty \gamma^t R_\phi(x_t, a_t, x_{t+1}; g)$, with $R_\phi(x_t, a_t, x_{t+1}; g) = \mathbb{1}\{x_{t+1} \in \phi(g)\}$ measures the optimality of actions taken by an agent with respect to achieving the goal $g$, where $\gamma \in [0, 1)$ is the discount factor penalizing long sequences of actions. We make no assumptions regarding the optimality of the data collection policy $\pi^\beta$ with respect to $\mathcal{G}^{tr}$ and thus, in what follows, we will view our pre-collected data as an un-ordered collection of state-action-state transitions, in short denoted as $\mathcal{D} = \{(x, a, x')\}$. We also do not assume access to either of the ground-truth state-transition dynamics $P$ or the goal-to-state mapping $\phi$, and consequently the reward $R_\phi$. We only require that $\mathcal{G}^{tr}$ contains goals corresponding to states that have been visited in $\mathcal{D}$.[2]

**The goal.** Given $\mathcal{D}$ and $\mathcal{G}^{tr}$, our objective is to learn a

---

[1] In this paper, we focus on simple goals representable as a subset of the state space. This can be extended to more complex goals using temporal logic which we leave for future work.

[2] In practice, $\mathcal{G}^{tr}$ can consist of a much larger set of training goals. This set will be effectively reduced to the set of visited goals after the first step of our training pipeline.

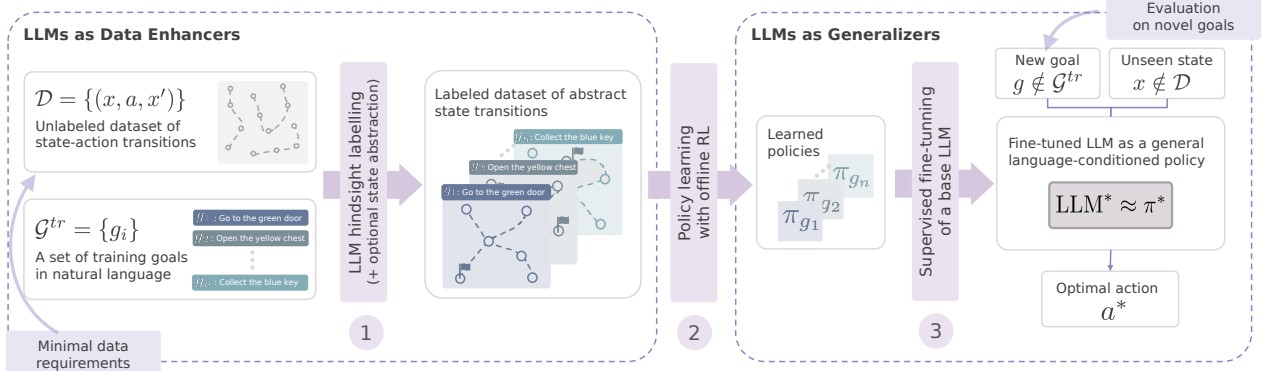

Figure 1. Overview of TEDUO. ① The unlabeled dataset of state-action transitions is pre-processed with LLM-automated hindsight labeling and state abstraction. ② The resulting labeled dataset of abstract state transitions is used as the input to an offline RL algorithm to learn the optimal goal-conditioned policies for the finite set of training goals. ③ Knowledge of the optimal actions for each observed training goal is distilled into a base LLM via SFT. The fine-tuned LLM acts as a language conditioned policy generalizing to previously unseen states and language commands.

language-conditioned policy $\pi^*$ maximizing the expectation of the cumulative discounted rewards averaged across all $g \in \mathcal{G}$. Crucially, $\pi^*$ should generalize to novel goals $g \notin \mathcal{G}^{tr}$ and previously unseen states $x \notin \mathcal{D}$. We require that $\pi^*$ not only generalizes to synonymous language commands, but also to previously unvisited goal-states.

# 3. The Method: TEDUO

To address the problem of learning language-conditioned policies solely based on the inputs $\mathcal{D}$ and $\mathcal{G}^{tr}$, we must overcome the challenges C1-C4 outlined in the introduction. While conventional RL methods are successful at learning optimal policies within well-explored environments, they typically require additional data labeling and are limited in generalization to new, previously unseen language commands and states. In contrast, although LLMs can understand the meaning of sentences in natural language describing each goal, their skills lack grounding in relation to the environment's dynamics. Our pipeline, TEDUO, employs LLMs to enhance conventional RL, effectively addressing challenges C1-C4. TEDUO consists of three main steps:

① **Construction of solvable MDPs.** For each goal, $g \in \mathcal{G}^{tr}$, we construct an MDP by employing LLM-automated hindsight labeling and optional state abstraction, addressing C1 and C2.

② **Offline Policy Learning.** After obtaining a labeled dataset for each goal in $g \in \mathcal{G}^{tr}$, we solve the set of abstract MDPs using an out-of-the-box offline RL algorithm. As a result, we obtain a set of learned policies $\{\pi_g : g \in \mathcal{G}^{tr}\}$. The learned policies improve on naive imitation learning, addressing C3.

③ **LLM supervised fine-tuning.** We distill the knowledge of the environment dynamics into a pre-trained LLM via SFT, grounding the prior knowledge of the base LLM

and thus, enabling generalization to new, previously unseen states and goals, addressing C2 and C4.

## 3.1. Step 1. Construction of solvable MDPs

Given that in many symbolic environments the original state representations $\mathcal{X}$ is high-dimensional, our pipeline contains an optional state abstraction step that maps raw observations $x \in \mathcal{X}$ to a goal-dependent abstract states $s^g \in \mathcal{S}^g$. For each $g \in G^{tr}$ we construct an MDP $\{\mathcal{M}^g : g \in \mathcal{G}^{tr}\}$, where $\mathcal{M}^g := (\mathcal{S}^g, \mathcal{A}, P^g, R^g, \rho, \gamma)$, with $P^g$ standing for the induced transition operator and $R^g$ the reward function with respect to the goal $g$ approximated with scalable LLM-based proxies. The state abstraction stage is entirely optional and if not employed, we simply take $\mathcal{S}^g = \mathcal{X}$ and $P^g = P$.

### 3.1.1. STATE ABSTRACTION (OPTIONAL)

The goal of state abstraction is to reduce the size of the state space by grouping together similar states in a way that reduces the complexity of the MDP (Li et al., 2006). With a well-designed abstraction, conventional RL algorithms can learn more efficiently, requiring fewer samples of data, which is particularly relevant in the offline setup. Let $F : \mathcal{X} \times \mathcal{G} \to \mathcal{S}^g$ be an abstraction operator that given a goal $g$ maps a single observation $x \in \mathcal{X}$ to an abstract state $s^g := F(x; g)$. This map should be such that $|\mathcal{S}^g| \ll |\mathcal{X}|$.

In the context of text-based environments, we can leverage natural language to guide state abstraction, so that the resulting abstract states contain only the goal-relevant information. Given an environment that can be essentially represented in a $d$-dimensional feature space, $\mathcal{X} = \mathcal{X}^1 \times \mathcal{X}^2 \times \ldots \times \mathcal{X}^d$, we assume that only a relatively small subset of these variables is relevant for solving a specific goal $g$ and an even smaller subset is required to identify the goal states $\phi(g)$. For details on the implementation of the LLM state abstrac-

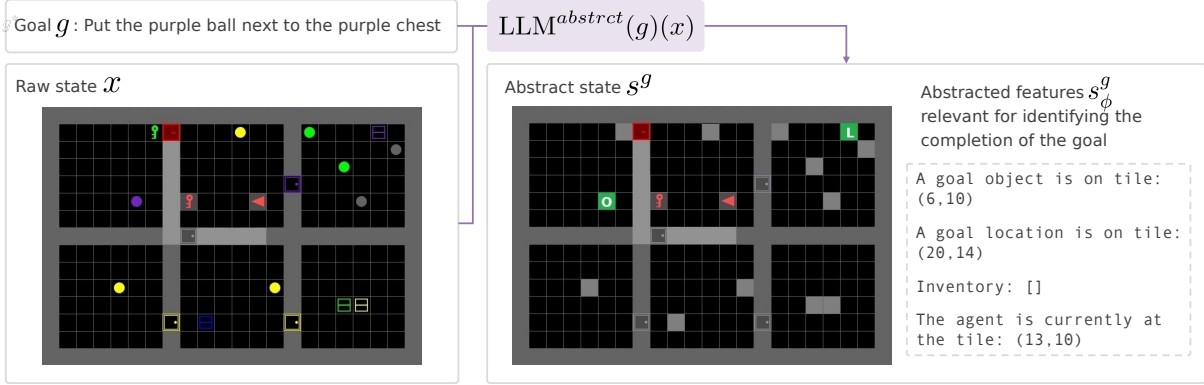

*Figure 2. An example of state abstraction for a grid world.* The LLM-induced abstraction function reduces the complexity of the original state by treating irrelevant distractors as walls, disregarding the color of opened doors, and identifying the object to be picked up (marked with "O") and its designated location (marked with "L").

tion operator refer to Appendix C.4. Figure 2 shows an example effect of state abstraction on a grid-world from the BabyAI environment.

### 3.1.2. GOAL-CONDITIONED HINDSIGHT LABELING.

Following recent works (Kwon et al., 2023), we hypothesize that the existing abilities of LLMs are sufficient to perform the simple task of identifying whether a particular state belongs to the set of goal states $\phi(g)$ associated with the goal description $g$. In order to perform hindsight labeling of our dataset $\mathcal{D}$, we wish to approximate $\phi$, and thus the reward $R_\phi(\cdot; g)$, with a prompted language model $\text{LLM}^{rwrd} \approx \mathbb{1}_{(g \text{ is achieved in } s^g)}$, where we assign the rewards in the abstracted spaces $\mathcal{S}^g$ instead of $\mathcal{X}$. Given the large number of goals and states, to reduce the number of LLM calls needed, instead of using language models directly, we train proxy reward models–lightweight neural networks $R_\theta(\cdot; g) : \mathcal{S}^g \to \{0, 1\}$–to predict the labels generated by the prompted language model, $\text{LLM}^{rwrd}$. Details can be found in the Appendix C.5. The proxy rewards functions provide a much more cost-effective way to perform hindsight labeling compared to labeling all states from $\mathcal{D}$ for all goals from $\mathcal{G}^{tr}$ directly by LLM prompting let alone by human annotators. Appendix B.5 shows that for the BabyAI environment proxy rewards reach near 100% accuracy in comparison to the ground truth rewards of the environment.

### 3.2. Step 2. Offline Policy Learning.

After the first step, for each goal $g \in \mathcal{G}^{tr}$ we obtain an offline dataset $\mathcal{D}^g := \{(s^g, a, s^g, r^g)\}$. Given these data, we can apply any offline reinforcement learning method to learn optimal policies $\pi^g$, for each goal $g \in \mathcal{G}^{tr}$. In practice, however, to learn the goal-conditioned policies, the chosen RL method should be scalable, as we must solve multiple MDPs, one for each goal in $\mathcal{G}^{tr}$. Therefore, in our instantiation, we discard computationally intensive methods. Furthermore, as the generalization to new states is tackled by the next

step, we do not require at this stage that the learned policies generalize to unseen states. Given these considerations, we primarily employ tabular Q-learning (Watkins & Dayan, 1992) to solve the set of abstract MDPs. To demonstrate the flexibility of our approach, we also conduct experiments with Deep Q-Learning (Appendix B.4) and filtered Behavioral Cloning (Appendix B.1). At the end of this stage, we obtain a set of learned policies $\{\pi^g : g \in \mathcal{G}^{tr}\}$. These policies are limited to the set of training goals and the set of states observed in $\mathcal{D}$.

### 3.3. Step 3. Training the LLM as a generalizable policy

To enable generalization to previously unseen states, and more importantly, generalization to novel goals, the final step of our method distills the knowledge of the optimal actions per each abstract state and goal into a pre-trained LLM. We build a supervised dataset $\mathcal{D}^{SFT}$ consisting of goal commands, initial abstract states and the sequence of optimal actions with respect to the learned policies. Concretely,

$$\mathcal{D}^{SFT} := \{(g, s_0^g, [a_0^{*,g}, \dots, a_{n_g}^{*,g}]) : g \in \mathcal{G}^{tr}, \ s_0^g \in \mathcal{D}^g,$$
$$a_t^{*,g} = \arg\max_{a \in \mathcal{A}} \pi^g(a \mid s_t^g),$$
$$s_{t+1}^g = \arg\max_{s \in \mathcal{S}^g} \hat{P}^g(s | s_t^g, a_t^{*,g})\},$$
$$n_g \ s.t. \ R_{\hat\theta}(s_{n_g+1}^g; g) = 1\},$$

where $\hat{P}^g$ is the empirical state transition function based on the abstract datasets $\mathcal{D}^g$, obtained in step 2. We then fine-tune a pre-trained large language model on $\mathcal{D}^{SFT}$ using the standard next-word prediction objective. We integrate description of the goal $g$ and the state $s_0^g$ into a prompt and set the sequence $[a_0^{*,g}, \dots, a_{n_g}^{*,g}]$ as the expected completion. We expect that the fine-tuned language model combined with the state abstraction function $\text{LLM}^{abstrct}$ can effectively act as a proxy for the general, goal-conditioned policy $\pi^*$, generalizing to any new goal $g \notin \mathcal{G}^{tr}$ and previously unobserved low-level state $x \in \mathcal{X}$.

## 4. Related Work

**LLMs for decision making.** There is growing interest in using general-purpose LLMs directly as decision-making agents (Yao et al., 2023b). Various prompting techniques, such as chain-of-thought (CoT) (Wei et al., 2023) and self-reflection (Ji et al., 2023), have been developed to enhance LLMs' abilities in long-term planning tasks. Yet, prompting alone is insufficient for solving complex tasks in dynamic environments (Szot et al., 2024; Finn, 2024). To effectively utilize the prior knowledge of LLMs, they must be grounded in the environment dynamics. This can be achieved either through in-context learning (Wang et al., 2023; Wu et al., 2023) or fine-tuning (Carta et al., 2023; Tan et al., 2024; Brohan et al., 2023a). A key limitation of in-context learning is its restricted window size. In this work, we focus on fine-tuning; however, unlike prior studies, we significantly reduce the requirements on the input data.

**LLMs as data enhancers.** In conventional goal-conditioned RL, datasets of state-action transitions require augmentation with goal-dependent rewards, often through human annotation or learning from demonstrations (Ziebart et al., 2008; Fu et al., 2018; Bahdanau et al., 2018). Recent studies show pre-trained LLMs can generate task-specific rewards (Yu et al., 2023b; Ma et al., 2023; Xie et al., 2023), though most rely on costly iterative prompting. We reduce these costs by approximating LLM-induced reward functions with proxy neural networks and assigning rewards in abstracted state spaces, lowering labeling needs. Similar to Peng et al. (2023), our approach uses natural language to guide the elimination of irrelevant state features.

**Language-conditioned RL.** Prior work on language-conditioned policies often assumes access to ground-truth rewards (Jiang et al., 2019; Co-Reyes et al., 2018), real-time experimentation (Fu et al., 2018; Bahdanau et al., 2018; Mirchandani et al., 2021), or expert demonstrations paired with language annotations (Stepputtis et al., 2020; Xiao et al., 2023; Brohan et al., 2023b;a). Our approach learns from offline, unlabeled datasets that are potentially suboptimal and lack reward signals. While most language-conditioned RL studies evaluate on synonymous commands seen during training (Lynch & Sermanet, 2021; Nair et al., 2022), we focus on novel instructions corresponding to previously unsolved goals which is a significantly more challenging setup that only a few related works attempt to address (Xiao et al., 2023; Brohan et al., 2023a; Jang et al., 2022).

See Appendix A for an extended discussion of related works.

## 5. Experiments

**Questions.** In our experiments we aim to answer the following questions: **(Q1)** Does the use of a pre-trained language model enable generalization to new language commands and new states? **(Q2)** How does our method compare to simpler prompting-based methods and alternative approaches to language-conditioned RL? **(Q3)** As a result of SFT, does the language model memorize the optimal actions or does it learn *generalizable* and *compositional* skills? **(Q4)** How does our method scale with computer power and what is the effect of language abstractions on data efficiency?

**Experimental Setup.** We require a controlled with environment where a wide variety of distinct goals can be specified, whose state-spaces can be represented in natural language. We choose the BabyAI (Chevalier-Boisvert et al., 2018) environment as our main benchmark and the Webshop environment (Yao et al., 2023a) as an additional environment with an increased complexity of goals and actions, demonstrating the adaptability of TEDUO beyond grid-world environments. A detailed discussion on these choices can be found in Appendix A. **BabyAI** is a grid world platform for instruction following where an agent receives natural language goal instructions such as: *"Go to the tile (3,2)"* or *"Look behind the green locked door"*. The grids can consist of multiple rooms connected by open or locked doors and different distractor objects that the agent can interact with (boxes, keys, balls, etc.). The action space $\mathcal{A}$ consists of several navigation primitives (`forward`, `pickup`, etc.). **Webshop** is a simulated e-commerce environment where an agent given product requirements must locate corresponding items by navigating a website. This environment is particularly relevant due to its *dynamically evolving action space*. The available actions vary by state due to changes in the website's UI elements. Furthermore, the agent must generate linguistic inputs (e.g., search queries) to use the search bar. This highlights the necessity for the fine-tuned LLM to avoid catastrophic forgetting (Luo et al., 2025) and produce coherent keywords for narrowing searches. We work with the simplified HTML representation provided in the environment as our abstracted state space. Due to space constraints, results for the Webshop environment are presented in the Appendix B.1.

**Metrics.** We rely on the following metrics to evaluate our learned policies: *success rate*: proportion of attempts in which the agent achieves the goal within the time limit (500 steps); *episode length*: the average number of steps taken to reach the goal or the time limit; *invalid actions*: ratio of invalid actions (e.g., moving into a wall) to total actions.

### 5.1. Q1: Online Evaluation: Generalization Benchmark

**Setup.** We choose the collection of `Synth` environments–the most complex environments not requiring state memory from BabyAI–as the main test bed for TEDUO. All environments are constructed as a 22x22 grid and containing 9 rooms. They differ in the type, position, and color of the distractors. The tasks include goals such as "go to the {color} {object}", "pick up the {color} {object}", or "put

*Table 1. Online evaluation of generalization performance.* Results averaged over 400 $(g, s_0^g)$ pairs

| Method | Environ-ment | Goals | Success Rate [%] | Episode Length | Invalid Actions [%] |
|---|---|---|---|---|---|
| Llama-3-8B (vanilla) | train/test | train/test | 17 (±0.9) | 444 (±3.2) | 42 (±0.1) |
| Llama-3-70B (vanilla) | train/test | train/test | 14 (±0.7) | 452 (±3.0) | 55 (±0.2) |
| Llama-3-8B (in-context+CoT) | train/test | train/test | 16 (±0.7) | 443 (±3.3) | 42 (±0.1) |
| Llama-3-70B (in-context+CoT) | train/test | train/test | 21 (±0.9) | 432 (±3.8) | 47 (±0.3) |
| DeepSeek-R1 (distilled to Llama-8B)[a] | train/test | train/test | 32 (±3.6) | 379 (±14.4) | 40 (±0.5) |
| TEDUO: steps 1 & 2 + BabyAI-IL-bot | train | train | 69 (±1.2) | 248 (±4.9) | 17 (±0.6) |
| | test | train | 45 (±1.2) | 344 (±4.8) | 19 (±0.6) |
| | train | test | 15 (±0.8) | 453 (±2.9) | 44 (±0.7) |
| | test | test | 16 (±0.8) | 447 (±3.1) | 36 (±0.6) |
| **TEDUO**-Llama-3-8B | train | train | 65 (±1.4) | 203 (±6.7) | 21 (±0.7) |
| | test | train | 53 (±1.1) | 257 (±5.4) | 27 (±0.7) |
| | train | test | 55 (±1.6) | 241 (±7.5) | 22 (±1.1) |
| | test | test | 45 (±1.3) | 286 (±6.1) | 31 (±1.2) |

[a] DeepSeek-R1 generates a reasoning trace per action, resulting in 100× longer responses by token count and inference time. Consequently, results are averaged over 50 $(g, s_0^g)$ pairs.

the {color} {object} next to the {color} {object}". A list of 500 goals is used as $\mathcal{G}^{tr}$. The set of testing goals contains 100 goals that are semantically distinct from those in $\mathcal{G}^{tr}$. The set of testing goals is augmented by asking GPT-4 to paraphrase the original commands provided by BabyAI. We train a Llama-3-8B model with TEDUO based on a dataset $\mathcal{D}$ containing 800k non-unique state-action-state triplets generated according to a policy that is a random mixture of default policies from BabyAI[3] (see Appendix C.3 for details) .

**Baselines.** We compare our fine-tuned Llama-3-8B agent with non-fine-tuned LLMs: DeepSeek-R1 (DeepSeek-AI, 2025), Llama-3-8B and Llama-3-70B (Llama Team, 2024) using a) vanilla and b) CoT prompting (Wei et al., 2023) with additional demonstrations provided in-context (in-context+CoT). The latter integrates expert demonstrations generated during step 2 of TEDUO to test the in-context learning ability of the LLM. Following recent works (Mezghani et al., 2023; Li et al., 2022; Cao et al., 2023), we also compare against BabyAI-IL-bot, the baseline proposed by the authors of BabyAI (Chevalier-Boisvert et al., 2018), which is the combination of a GRU to encode the instruction, CNN+FILM layers to encode the grid and an LSTM memory. We train this method via imitation learning on the policy generated by TEDUO, steps 1&2. Implementation details of can be found in Appendix C.8.1.

**Results.** Based on the results presented in Table 1 we make the following observations:

**Prior knowledge of LLMs is insufficient.** We find that non-fine-tuned LLMs, irrespective of their parameter count or prompting method struggle in solving the seemingly simple tasks from the BabyAI environments. Low success rate

and high invalid action ratios indicate the inability of LLMs to understand the dynamics of the environment. Common failures include only using the action "move forward" without considering the agent's direction or attempting final actions (e.g. door opening) without first navigating to the correct location. This underscores the need for developing data-efficient methods for distilling knowledge of the environment-dynamics into LLMs. Our fine-tunning strategy brings the success rate from 17% to 65% in its training setting and 45% for testing on novel goals.

**Unlocking generalization.** We evaluate the generalization abilities of the fine-tuned TEDUO-Llama-3-8B model to new environments and goals. When tested on new environments unseen during training, a performance drop of 12% is observed, significantly smaller than BabyAI-IL-bot with a drop of 24%. This can be explained by the overfitting of the baseline due to the limited offline training data. TEDUO-Llama-3-8B benefits from the zero-shot capabilities of the pre-trained LLM. This effect is even more pronounced with new goals–TEDUO experiences only an 8% decrease in success rate, compared to the 40% drop for the BabyAI-IL-bot. Overall, TEDUO achieves nearly three times better performance than the RL baseline when generalizing to both new natural language commands and environments. Appendix B.6 shows success rates by goal type. Additional experiments demonstrating TEDUO's generalization to larger, unseen BabyAI grids after training on smaller grids are presented in Appendix B.7. Performance remains robust for grids three times larger than those seen during training, confirming TEDUO's generalization ability.

### 5.2. Q2: Online Evaluation: Ablation Study

**Setup.** With the same experimental setup, we compare our full fine-tuning pipeline with its ablations. After obtaining the abstract datasets $\mathcal{D}^g$ with the first step of TEDUO, we

---

[3]Codebase to reproduce the main results of this paper can be found at this link.

*Table 2. Ablation study.* Results averaged over 400 $(g, s_0^g)$ pairs.

| Method | Success Rate [%] | Episode Length | Invalid Actions [%] |
|---|---|---|---|
| Step 1 + GCBC | 7 (±0.6) | 474 (±2.3) | 11 (±0.1) |
| Steps 1+2 (GCRL) | 16 (±0.8) | 430 (±3.9) | 10 (±0.1) |
| All steps Llama-3-8B | 65 (±1.4) | 203 (±6.7) | 21 (±0.7) |

generate goal-conditioned policies with naive behavioral cloning (step 1 + GCBC). We also compare our fine-tuned Llama-3-8B against the performance of the GCRL policies obtained with offline Q-learning in step 2. Note, neither of GCBC nor GCRL can generalize to new, previously unseen language commands. Therefore, in this study, we are only looking at performance on goals from $\mathcal{G}^{tr}$. Ablation of the abstraction function is delayed to the next section.

**Results.** The results of GCBC and GCRL can be seen as ablations of our pipeline. We first note that the success rate of naive behavioral cloning is low, indicating low fidelity of the data collection policy and highlighting the need for incorporating offline policy-learning methods. Moreover, the significantly improved performance of the Q-learning policies (GCRL) validates the effectiveness of the first two steps within our pipeline. The synthetically constructed MDPs are meaningful offline constructs that yield policies effective during online testing. Finally, the improved performance of our fine-tuned Llama-3-8B over the Q-learning/GCRL policies on training goals and environments confirms the importance of the third step of our method and suggests that the ungrounded, prior knowledge of large language models improves generalization to new previously unseen states.

### 5.3. Q3: Learning and exploiting core skills

We wish to investigate if by learning the optimal policies for diverse goals and environments, the LLM can integrate core skills required to achieve these goals and whether such skills can be transferred across tasks. We also investigate the aspect of skill compositionality. Does the prior knowledge of the LLM, now grounded in the environment dynamics, suffice to compose together learned skills?

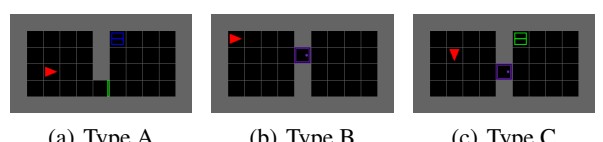

(a) Type A     (b) Type B     (c) Type C

*Figure 3. Three simple environment types.*

#### 5.3.1. SKILL TRANSFER AND COMPOSITIONALITY.

**Setup.** We are working with three types of simple environments illustrated in Figure 3. The position and color of the door and box vary across different instantiations of the environments. We use type A and B environments for

*Table 3. Performance on type C test tasks.* TEDUO A and TEDUO B have been trained in only one environment whereas TEDUO A&B has been trained in both.

| Method | Success Rate [%] | Episode Length | Invalid Actions [%] |
|---|---|---|---|
| LLM (vanilla) | 0 (±0.0) | 20 (±0.0) | 75 (±0.5) |
| TEDUO A | 0 (±0.0) | 20 (±0.0) | 51 (±0.7) |
| TEDUO B | 0 (±0.0) | 20 (±0.0) | 28 (±0.4) |
| TEDUO A&B | 60 (±2.1) | 16 (±0.2) | 37 (±1.0) |

training and type C environments for testing. We note that tasks from type C environments require the internalization of three core skills: moving to a given location, opening a door, picking up a box. The skill of moving to a location can be obtained from both environments A and B, but the skill of picking up a box or opening the door can only be obtained from one of the environments, A or B, respectively. This setup allows us to investigate the transferability of learned skills across environments and their compositionality.

**Results.** Table 3 shows that agents trained on one type of environments only fail to generalize. TEDUO A reaches a 99% success rate in tasks without closed doors (type A grids), but consistently fails when the goal is behind a door. TEDUO B achieves an 81% success rate in new grids from Type B but cannot generalize to Type C. The fact that TEDUO A&B can generalize to the environment C that requires a combination of both skills independently seen during training with a high success rate of 60% indicates that the fine-tuned LLM does not merely memorize optimal trajectories for individual tasks. Instead, it learns core, generalizable abilities that can be combined to solve novel tasks. This result emphasizes the significance of multi-skill learning for successful generalization that TEDUO enables.

#### 5.3.2. INTERNALIZATION OF CORE SKILLS.

One of the core skills required to successfully solve the tasks is to identify whether the agent at its location is facing an object or a wall, or it is free to move forwards. This section provides additional insights into the behavior and internal state representation of the LLM fine-tuned with TEDUO in comparison to a base LLM.

**Setup.** As in the main evaluation benchmark, we operate within the Synth environments and generate a dataset with 10 random goals and 512 states per each goal. We embed each goal-state pair into the prompt template for eliciting actions and LLM fine-tuning and pass them through both the base and fine-tuned Llama-3-8B from experiments 5.1 and 5.2. We record the logprobabilities of the tokens [0, 1, ..., 6] as well as the hidden representation of states at each layer. We label our dataset according to whether at the given state the agent is facing a wall or an object. For each layer, we fit two linear probes on top of the hidden representations: one for wall and one for object detection.

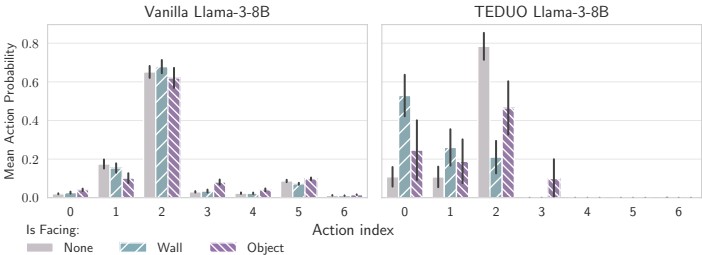
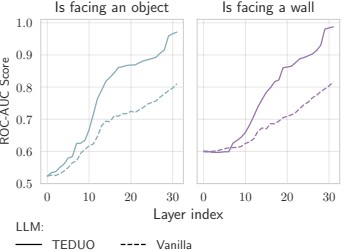

(a) *Action probabilities.* Action codes: 0: left, 1: right, 2: forward, 3: pick up an object, 4: drop an object, 5: toggle an object, 6: done

(b) *ROC-AUC score of the linear probe.*

*Figure 4. Interpretability results for detection of walls and objects.*

**Results.** First, from Figure 4(a), we observe that a non-fine-tuned Llama-3-8B puts a high probability on the action 'move forwards' irrespective of whether the agent is facing an obstacle or not; this results in a high ratio of invalid actions, as previously observed in the benchmark experiments. After fine-tuning with TEDUO, the probability of moving forwards when facing an obstacle is significantly reduced, putting more weight on the actions of moving left or right to avoid the obstacle. We also observe, that our TEDUO method taught the LLM that objects can be picked-up (action 3), only when the agent is directly facing it. From the linear probe experiments (Figure 4(b)) we observe that after fine-tunning, the internal representations of states directly encode the information of whether the agent is facing an obstacle. At the final layers, the ROC-AUC score of predicting both types of labels is near 100%, in sharp contrast with the score of around 80% for the non-fine-tuned model. Yet, the score of 80% is still relatively, high, indicating that the original state representations are sufficient to identify whether the agents is facing an obstacle, but, since the non-fine-tuned LLM lacks grounding of this knowledge with respect to the environment dynamics, it struggles to translate it into an optimal action to be taken. This result underscores the claims of previous works that out-of-the-box LLMs struggle to translate their prior knowledge into low-level actions within dynamic environments (Finn, 2024; Szot et al., 2024).

### 5.4. Q4: Data efficiency and Scaling of TEDUO

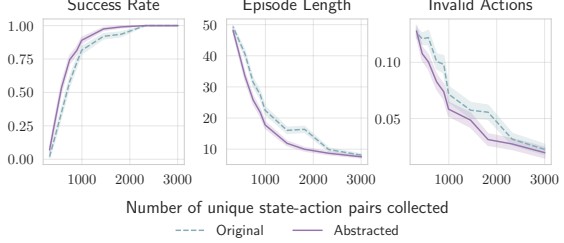

*Figure 5. Performance vs. offline dataset size.* The abstraction function enhances data efficiency.

**Impact of state abstraction.** Figure 5 shows the performance of the Q-learning policies (i.e., policies $\pi_g$ obtained at the end of TEDUO step 1+2) against the size of the observational dataset $\mathcal{D}$. This experiment is realized with and

without the abstraction function during step 1. As anticipated, the efficacy of the learned policies improves with increasing size of the dataset $\mathcal{D}$. On average, the LLM-based state abstraction reduces the number of unique states by 10% (see Fig. B.2). The reduction in state space size significantly enhances data efficiency of our training method across all three performance metrics. Furthermore, the size of the state spaces $\mathcal{S}_\phi^g$, corresponding to the subset of features relevant for identifying the completion of the goal is reduced to just around 20% of the original state space size (Fig. B.2 in the Appendix). This reduces the size of the datasets used for training and the subsequent goal-identification 5-fold.

**Compute power is the new bottleneck.** Given a fixed observational dataset $\mathcal{D}$, we can expand at no extra cost the fine-tuning dataset $\mathcal{D}^{SFT}$ by introducing more training goals in $\mathcal{G}^{tr}$. Yet, larger $\mathcal{D}^{SFT}$ necessitates more compute power for training the LLM agent. Table 4 demonstrates the

*Table 4. Performance vs. compute power.*

| TFlops | $|\mathcal{G}^{tr}|$ | Success Rate [%] | Episode Length | Invalid Actions [%] |
|---|---|---|---|---|
| 5.2e7 | 266 | 33 | 342 | 32 |
| 8.6e7 | 372 | 36 | 330 | 40 |
| 1.4e8 | 534 | 45 | 286 | 31 |

scaling of our method with compute power. As expected, training on a wider range of goals results in an improved performance on unseen test goals. We do not observe a plateau in performance metrics, suggesting that with additional compute further gains may be possible. Consequently, our approach shifts the bottleneck from the limited availability of real observational data to computational power.

## 6. Discussion

**Limitations.** Leveraging LLMs' prior knowledge enables efficient policy generation with minimal data. However, some applications may benefit more than others. First, certain scenarios may be out of distribution even for LLMs trained on extensive Internet data. Second, we assume that the environment state can be represented textually, which, although feasible for many applications due to language's expressiveness, may not be ideal in all cases (this can be

owever mitigated by employing VLMs which we leave as future work). Third, due to the discrete nature of LLM tokenization, using fine-tuned LLMs to directly output actions requires discretization of the action space, which can hinder performance in continuous control tasks. Lastly, while data requirements are minimal, they still assume some practitioner knowledge of the environment and the data $\mathcal{D}$ to propose the list of goals $\mathcal{G}$ (see Appendix B.2 for details).

**Conclusions.** We introduced a novel framework for training natural language instruction-following agents capable of generalizing to new states and instructions in a zero-shot setting. This is the first method to learn natural language goal-conditioned policies in an offline setting using observational data that is both unlabeled and non-expert. The success of TEDUO relies on the synergy between the mutual strengths of conventional RL and LLMs. Given the demonstrated flexibility of this method in generalizing to new goals and environments, future work could explore its potential in learning across multiple environments with distinct action and state spaces.

## Impact Statement

This paper presents work whose goal is to advance the field of Machine Learning. There are many potential societal consequences of our work, none which we feel must be specifically highlighted here.

## Acknowledgments

This work was supported by Azure sponsorship credits granted by Microsoft's AI for Good Research Lab. Thomas Pouplin's research is supported by funding from AstraZeneca. Katarzyna Kobalczyk's research is supported by funding from Eedi. Hao Sun's research is supported by funding from the Office of Naval Research (ONR).

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

# A. Extended Related Work

## A.1. Generalization in offline Reinforcement Learning

Following the work of Mediratta et al. (2024), we separate the generalization abilities of offline reinforcement learning algorithms into two categories: new instruction following and adaptation to new states or environments.

**Goal-conditioned RL.** Goal-conditioned Reinforcement Learning (GCRL) is a subfield of RL dedicated to developing policies capable of achieving multiple goals within the same environment dynamics. These policies are conditioned on an additional input, $g$, indicating the goal that the next action should aim to achieve. While most recent research has focused on online settings (Islam et al., 2022; Han et al., 2021; Hong et al., 2023; Yang et al., 2021), only a few methods have addressed the offline GCRL problem (Yang et al., 2022; Ma et al., 2022; Chebotar et al., 2021). (Yang et al., 2023) offers a comparison of these methods and highlights the key challenges in offline GCRL. Additionally, these approaches typically restrict goal representations to those expressible as a single state in the state space (Chebotar et al., 2021), a scalar parameter (Ma et al., 2022), or a fixed set of known goals (Yang et al., 2022).

**Language-conditioned RL.** Our work addresses the problem of goal-conditioned RL, where goals are expressed in natural language. While using language to specify goals is natural and broadens the range of possible goals it comes with the challenge of grounding the semantics of language in the environment state space and dynamics. Such language-instruction following agents have been widely studied in both reinforcement learning and imitation learning contexts. However, most existing methods either rely on access to an online environment for interaction (Fu et al., 2018; Bahdanau et al., 2018; Jiang et al., 2019; Mirchandani et al., 2021) or require costly, goal-annotated expert datasets of offline demonstrations (Stepputtis et al., 2020; Brohan et al., 2023b;a). In contrast, our approach does not assume any environment-provided reward signal or access to real-time exploration. Furthermore, in terms of generalization to new natural language instructions, we distinguish between evaluation on instructions that simply paraphrase the training goals (Nair et al., 2022; Lynch & Sermanet, 2021) from those that represent semantically novel goals. Similar to the works of Xiao et al. (2023); Brohan et al. (2023a); Stepputtis et al. (2020); Shridhar et al. (2021a); Jang et al. (2022), our focus is on the latter, more challenging scenario.

**Domain Generalization.** While the previous section addressed generalization to new goals, this section focuses on generalization to novel state-action transitions. This type of generalization extends beyond goal-conditioned RL, as it is essential even for single-goal RL. It has been widely studied and observed that Offline RL methods often overfit to the training distribution of state-action transitions, resulting in poor performance when the test distribution differs. Various approaches have been proposed to address this distribution shift, including regularization techniques (Kostrikov et al., 2021; Kumar et al., 2020), model-based RL (Yu et al., 2020; Kidambi et al., 2021), and enhanced representation learning (Mazoure et al., 2021; Fan & Li, 2022). In TEDUO, we intentionally avoid domain generalization when solving the abstract MDPs in step 2 to prevent providing incorrect examples to the LLM in step 3. However, our method achieves domain generalization by leveraging the zero-shot capabilities of the fine-tuned LLM. Future work could enhance TEDUO by replacing tabular Q-learning in step 2 with a method that generalizes to new state-action transitions.

## A.2. Offline policy learning with minimal data requirements.

This paper focuses on realistic requirements regarding the training inputs. We work under offline setting, with a limited number of unlabeled environment transitions (i.e., $(x_t, a_t, x_{t+1})$ triplets) and without any assumptions about the policy that generated the actions. To address this scenario, we employ LLMs to label the data, enabling the use of RL methods, and as an abstraction function to enhance sample efficiency. Below we discuss the related work regarding these two steps.

**Hindsight labeling**. Labeling data for goal-conditioned RL requires the design of reward functions for each goal. The most common approach for designing the rewards relies on handcrafted methods that are often require multiple refinements through trial and error (Knox et al., 2022). With a large number of goals, manual reward design becomes infeasible. Inverse Reinforcement Learning (Ziebart et al., 2008; Fu et al., 2018) attempts to generate reward functions directly from data, but it requires a large amount of expert demonstrations. Recent studies have explored the use of LLMs and VLMs as reward functions. These methods typically involve creating a preference dataset (Klissarov et al., 2023), comparing the cosine similarity between natural language goals and state representations, or leveraging the coding abilities of LLMs (Yu et al., 2023a), especially in an online iterative fashion (Ma et al., 2023; Xie et al., 2024). These approaches are however aimed at densifying the reward signal. In contrast, our method requires generating reward labels for a large number of goals (approx. 100-1000), making the scalability of the process crucial. Therefore, we focus on generating rewards with a limited number

of LLM calls. Our approach relies on LLM-based detection of task completion, which has been proven effective by (Kwon et al., 2023). We further reduce the number of LLM calls by approximating the LLM-generated rewards with lightweight proxy neural networks.

**State abstraction.** State abstraction aims to reduce the complexity of the state space by eliminating irrelevant features, thereby improving the efficiency of learning algorithms. Early work in this area focused on state aggregation, where similar states are grouped together to form more compact representations, with state similarity defined through the transition dynamics, value- or Q-functions (Andre & Russell, 2002; Li et al., 2006; Givan et al., 2003; Abel et al., 2018). Recent advancements have explored more sophisticated methods, such as deep learning-based state abstractions, employing neural networks to learn abstract representations of states (Allen et al., 2021). In this work, we explore the use of LLMs to accomplish the task of state abstraction. Our approach relies on prompting a pre-trained LLM to remove the features of a state that are irrelevant in solving the given goal. Such LLM-based state abstraction has been previously shown effective in the context of robotics by Peng et al. (2023) who employ LLMs to translate the language command into a binary mask highlighting the location of the goal-object.

### A.3. Large Language Models for Decision Making

**Decision Transformers.** Pre-trained models based on the Transformer architecture have been widely used to address decision-making problems. However, this paper does not focus on Decision Transformer (DT) models (Chen et al., 2021). Although DTs have been applied in goal-conditioned RL and IL (Xu et al., 2022; Raparthy et al., 2023; Putterman et al., 2022), the joint modelling of goal, state, and action representations remains challenging and requires large labeled datasets. Instead of training a decision transformer, this papers leverages the prior knowledge accumulated in LLMs trained on Internet data to a) enable effective use of the limited offline, unlabeled data, b) enable generalization to previously unseen goals and states.

**General-purpose LLMs for decision making.** Utilizing off-the-shelf LLMs has gained significant attention due to its simplicity. In decision-making, LLMs have been used to create assistance functions within training pipelines to enrich data (Klissarov et al., 2023; Yu et al., 2023a; Ma et al., 2023; Xie et al., 2023; Laskin et al., 2022), and as high-level planners during inference to guide traditional RL policies (Shah et al., 2023; Ahn et al., 2022). Additionally, there has been growing interest in using general-purpose LLMs directly as decision-making agents (Yao et al., 2023b). Improving the reasoning capabilities of LLM agents is now an active research area, focusing on methods that are independent of traditional RL. These include iterative prompting techniques such as self-reflection (Ji et al., 2023), CoT reasoning (Wei et al., 2023), and integration with planning algorithms like Monte Carlo Tree Search (Pouplin et al., 2024). Nevertheless, such methods have been shown inefficient in completing complex, multi-step decision-making tasks in dynamic environments (Finn, 2024; Szot et al., 2024). To effectively use the knowledge embedded in LLMs for solving RL problems, these models need to be grounded in the dynamics of the environment.

**Grounding LLMs with the environment dynamics.** An LLM agent grounded in an environment can link the semantics of both observations and possible actions to its internal representation system, enabling appropriate decision-making (Carta et al., 2023; Harnad, 1990). One approach to achieve such grounding is through **in-context learning**. For instance, Voyager (Wang et al., 2023) pushes the concept of an LLM agent to its limits by developing an automatic curriculum for GPT-4, supported by a library of executable programs, to play Minecraft. Another method involves providing the LLM with a game manual (Wu et al., 2023). However, these approaches either rely on extensive expert knowledge, such as carefully designed prompts, or on game manuals, which may not always be available. Additionally, in-context learning has limitations in data-driven scenarios, partly due to the restricted context window size, which is insufficient for incorporating entire datasets. An alternative approach involves **fine-tuning** LLMs to achieve grounding. Studies such as (Tan et al., 2024) and (Carta et al., 2023) use Proximal Policy Optimization (PPO) (Schulman et al., 2017) to propose online fine-tuning of LLMs. In the robotics domain, RT2 (Brohan et al., 2023a) demonstrates that co-fine-tuning on both web-scale data and expert robot demonstrations improves performance of VLMs for decision making in the context of robotics. Our method differs from previous work by significantly lowering the requirements on input data, as we do not need online interaction or labeled expert demonstrations. Furthermore, while RT2 implements co-fine-tuning, our method utilizes an off-the-shelf pre-trained LLM, which is then fine-tuned.

# B. Additional Results

### B.1. Webshop results

**Main Results.** The results averaged over 200 instructions are shown in Table B.1. Unlike Minigrid, Webshop does not feature multiple environment types, prohibiting generalization evaluation across environments. The scores represent the true environment reward (scaled to [0, 100]).

*Table B.1. Online evaluation of generalization performance for the Webshop environment.* Results averaged over 200 instructions.

| Method | Goals | Score | Episode Length |
|---|---|---|---|
| ReAct-Llama-3-8B | Training/Testing | 8.4 (±0.8) | 14.1 (±0.1) |
| ReAct-Llama-3-70B | Training/Testing | 13.8 (±0.9) | 14.2 (±0.1) |
| TEDUO-Llama-3-8B | Training | 52.1 (±0.6) | 6.6 (±0.1) |
| TEDUO-Llama-3-8B | Testing | 44.4 (±0.6) | 6.9 (±0.1) |

This benchmark demonstrates that TEDUO successfully learns from unlabeled trajectories in the Webshop environment. TEDUO significantly outperforms the ReAct prompting approach, particularly when applied to smaller models (compared to GPT-3.5 in the original paper). While ReAct prompting provides some improvement over standard prompting, its performance remains limited, whereas TEDUO achieves substantially better results.

*Table B.2. Ablation study for the Webshop environment.* Results averaged over 200 instructions.

| Method | Goals | Score | Episode Length |
|---|---|---|---|
| Data collection policy (LLM-random) | Training/Testing | 5.6 (±0.5) | 12.3 (±0.2) |
| Step 1 2 (GCRL) | Training | 56.2 (±0.6) | 6.9 (±0.1) |
| TEDUO-Llama-3-8B | Training | 52.1 (±0.6) | 6.6 (±0.1) |
| TEDUO-Llama-3-8B | Testing | 44.4 (±0.6) | 6.9 (±0.1) |

**Ablation study:** Similar to the approach in the paper, table B.2 provides an ablation study below that summarizes the performance improvements achieved after each TEDUO step. The results demonstrate that TEDUO Steps 1 & 2 effectively produce improved policies over the data collection policy for the training goals. Furthermore, TEDUO Step 3 successfully achieves its generalization objective, extending the learned policy's performance to new instructions. Compared to the BabyAI environment, the fine-tuned LLM does not outperform the Q-learning policies trained on the training instructions. This limitation can be attributed to the lack of diversity in the initial states. In the BabyAI environment, agents can start from any grid position, including configurations unknown to the tabular Q-learning policies, often resulting in failure trajectories. In contrast, the fine-tuned LLM demonstrates successful generalization in such cases. However, the webshop environment presents a single initial state—the search bar webpage. To surpass the Q-learning policies in this setting, the LLM would need to contradict its learned behaviors, making superior performance unattainable.

## B.2. The impact of the data collection policy

We examine the effect of the data collection policy on our pipeline's performance. Specifically, we demonstrate that our pipeline remains effective irrespective of the optimality of the data collection policy with respect to the set $\mathcal{G}^{tr}$.

Given the observational dataset $\mathcal{D}$ collected under a policy $\pi^\beta$, let $\mathcal{G}^\mathcal{D}$ represent the set of goals corresponding to goal states that have been visited in $\mathcal{D}$. This set is defined as:

$$\mathcal{G}^\mathcal{D} = \{g \in \mathcal{G} : \exists (x, a, x') \in \mathcal{D} \text{ s.t. } R_\phi(x, a, x'; g) = 1\}. \tag{1}$$

We can measure the alignment between the dataset $\mathcal{D}$ and the training goals $\mathcal{G}^{tr}$ by the size of $\mathcal{G}^\mathcal{D} \cap \mathcal{G}^{tr}$, i.e. the set of goals from $\mathcal{G}^{tr}$ that have been visited in $\mathcal{D}$. A key point is that, in step 2 of TEDUO, we cannot generate a policy $\pi^g$ for any goal $g$ not present in $\mathcal{G}^\mathcal{D}$. As discussed in section 5.4, the performance of the fine-tuned LLM depends on the size of the synthetically generated dataset $\mathcal{D}^{SFT}$, making $|\mathcal{G}^\mathcal{D} \cap \mathcal{G}^{tr}|$ an import ant metric for evaluating the fidelity of our training inputs: $\mathcal{D}$ and $\mathcal{G}^{tr}$.

To empirically analyze this, we consider two randomized policies:

- **A) Goal-oriented policy:** This is the policy used for data collection in the main experimental section. For each trajectory, a random goal from a set of goals $\mathcal{G}^\pi$ is drawn and the agents acts according to the goal-oriented policy provided in the BabyAI environment in order to achieve it. This policy simulates agents attempting to accomplish multiple task within the environment. Examples of real-world unlabeled data that could be generated from such policy include CCTV footage of employees at work, logs of medical procedures performed on a patient, or YouTube videos.

- **B) Random policy:** Actions are drawn uniformly at random from the action space. This policy represents an agent that explores the environment without a specific goal. Although this scenario is less common in real-world settings—where agents typically pursue objectives—it remains applicable to batch RL, particularly when learning from untrained agents with no prior knowledge.

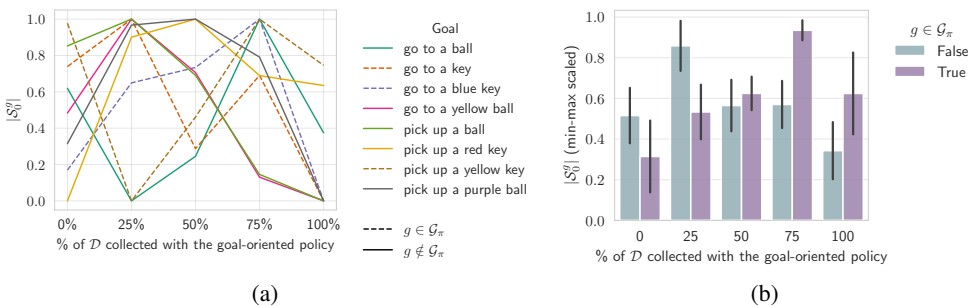

(a)                                                                 (b)

*Figure B.1. Impact of data collection policy.* The x-axis shows the proportion of data $\mathcal{D}$ collected with policy A vs. policy B for a fixed size of $\mathcal{D}$. a) The y-axis shows $|\mathcal{S}_0^g|$, i.e. the number of unique initial abstract states $s_0^g$ for which $g$ is reachable with the learned policy $\pi^g$. Values are min-max normalized across all 5 mixture policies. b) The y-axis shows the same values as plot a), averaged across 14 goals, bars represent the standard error.

**The metric of fidelity.** The likelihood of the final fine-tuned LLM in solving a given goal $g$ from any starting position is directly linked to the fidelity of the corresponding Q-learning policy, $\pi^g$. Thus, we chose to measure the fidelity of our starting point inputs by computing the number of initial abstract states $s_0^g$, from which $g$ can be solved using $\pi^g$. We define a goal $g$ as reachable from $s_0^g$ if by iteratively selecting the actions $a_t^* = \arg\max \pi^g(\cdot|s_t^g)$, the agent can eventually reach $g$. We denote this number of states with $|\mathcal{S}_0^g|$. The set of candidate initial states to compute $|\mathcal{S}_0^g|$ is the entire abstract state space.

Figure B.1 illustrates that the optimal data collection policy varies by goal. For some goals policy A works better, while for others it is the fully random policy. Importantly, a comparable amount of synthetic action sequences for fine-tunning the LLM can be extracted using either policy A or B. Averaging across all goals, we find that policy A tends to perform better for goals in $\mathcal{G}_\pi$ than those not in $\mathcal{G}_\pi$.

**Explanation:** We note that only a subset of initial states $s_0^g$ has been used during data collection. Using a goal-oriented policy to collect trajectories results in limited exploration of the state space. Consequently, the learned policies $\pi^g$ are expected to excel at solving $g$ when started from a state that has been visited in $\mathcal{D}$, but are not guaranteed to succeed when started from $s_0^g$ not visited in $\mathcal{D}$. This is why, including more exploration during data collection can help in ensuring that the learned policies can reach their respective goals from anywhere in the state space. However, relying solely on random exploration is only effective for simple goals that are likely to be reached by chance. Instead, in our main experiments, we use a goal-oriented collection policy that uniformly samples across goals of varying difficulty. This results in a more diverse dataset that includes complex behaviors and better reflects realistic settings, where completely random agents don't exist.

Future work could explore optimizing the set of training goals $\mathcal{G}^{tr}$ to maximize the alignment of $\mathcal{G}^{tr}$ with a given dataset $\mathcal{D}$. Yet, the necessity of aligning $\mathcal{D}$ and $\mathcal{G}^{tr}$ is moderated by two factors. First, as shown in subsection 5.4, the abstraction function reduces the complexity of the abstract MDPs, requiring fewer data samples. Second, since extending the list of goals in $\mathcal{G}^{tr}$ is computationally inexpensive, we can continually seek better alignment.

## B.3. Abstraction function

Figure B.2 presents the performance of the abstraction function used in the BabyAI environment.

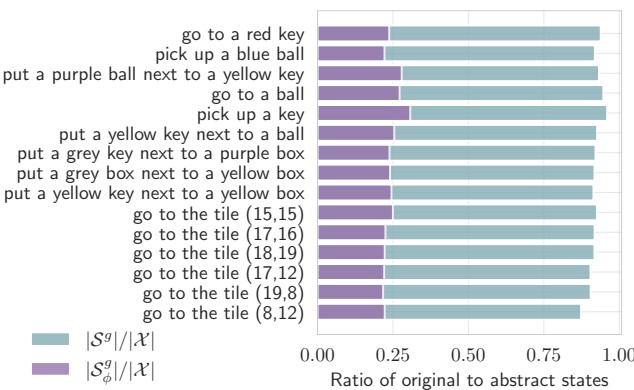

*Figure B.2.* Reduction in count of unique states due to applying the $\mathrm{LLM}(g)$ abstraction functions and the relative size of the reduced abstract feature space $\mathcal{S}^g_\phi$, containing only features necessary to identify the completion of a goal $g$.

## B.4. TEDUO step 2: Deep Q-Learning.

The step 2 of the TEDUO framework is independent of the specific offline reinforcement learning algorithm used to solve the abstract MDPs generated in step 1. Each abstract MDP is represented by a labeled transition dataset $\mathcal{D}^g$, consisting of tuples $(s^g, a, s'^g, r^g)$, making it compatible with any standard offline RL algorithm. In the implementation presented in Section 3, we used tabular Q-learning as it was most suitable for the environment under consideration. However, to illustrate the scalability of the approach, we show in table B.3 the performance of Deep Q-learning (DQN) (Mnih et al., 2013), demonstrating that it can effectively replace tabular Q-learning in our framework. The deep learning models for Q-value estimation combine CNN and dense layers, consistent with the lightweight models used for reward shaping (see Appendix C.5).While tabular Q-learning is limited to discrete state and action spaces and becomes impractical as their dimensions grow, DQN handles continuous spaces and scales effectively without such constraints.

*Table B.3. Ablation study with DQN.* Results averaged over 400 $(g, s^g_0)$ pairs.

| Method | Success Rate [%] | Episode Length | Invalid Actions [%] |
|---|---|---|---|
| Step 1 + GCBC | 7 (±0.6) | 474 (±2.3) | 11 (±0.1) |
| Steps 1 & 2 (GCRL- Tabular) | 16 (±0.8) | 430 (±3.9) | 10 (±0.1) |
| Steps 1 & 2 (GCRL- DQN) | 15 (±1.2) | 446 (±5.9) | 15 (±0.1) |
| All steps Llama-3-8B | 65 (±1.4) | 203 (±6.7) | 21 (±0.7) |

## B.5. Reward shaping evaluation

This section evaluates the performance of the reward-shaping step. We utilize pre-trained LLMs to identify states where a specific goal $g$ is achieved. As discussed in Section 3.1.2, the large number of states (around 200k) for each goal makes direct LLM usage impractical due to computational constraints. Therefore, the process is divided into two steps: (a) constructing a supervised dataset by labelling a subset of states (5k) using an LLM, and (b) training a lightweight neural network $R_\theta(\cdot; g)$ on this dataset.

*Table B.4. Reward Shaping Benchmark.* The accuracy, precision, and recall metrics are computed with a classification threshold ensuring at least 95% precision.

| Goals | ROC-AUG | Accuracy (%) | Precision (%) | Recall (%) |
|---|---|---|---|---|
| Go to a box | 0.90 | 89 | 96 | 38 |
| Pick up a ball | 0.75 | 98 | 95 | 83 |
| Open a door | 0.92 | 85 | 95 | 85 |
| Go to red door | 0.98 | 94 | 100 | 0.2 |
| Go to the tile (5,6) | 1.0 | 100 | 100 | 100 |
| Put a box next to a blue ball | 0.64 | 100 | 100 | 25 |

Table B.4 shows the performance of $R_\theta(\cdot; g)$ for various types of goals compared to ground truth rewards. The benchmark setup is consistent with the main experiments; details are provided in the Appendix C.5. All goals achieve 95% precision, a crucial metric since false positives lead to generating incorrect data points for $\mathcal{D}^{SFT}$ in TEDUO's step 2. Conversely, false negatives only reduce data points in $\mathcal{D}^{SFT}$, which is less critical given our synthetic data abundance (see Section 5.4). Performance varies across goals; for instance, "go to the red door" has low recall (0.2%), likely due to limited positive examples in the dataset. Expanding the dataset could improve such outcomes.

*Table B.5. LLM-only Reward Shaping Benchmark.* Accuracy, precision, and recall are computed with respect to LLM-generated labels.

| Goals | Accuracy (%) | Precision (%) | Recall (%) |
|---|---|---|---|
| Go to a box | 100 | 100 | 100 |
| Pick up a ball | 100 | 100 | 100 |
| Open a door | 100 | 100 | 100 |
| Go to a green door | 99 | 88 | 100 |
| Go to the tile (5,6) | 100 | 100 | 100 |
| Put a box next to a blue ball | 100 | 100 | 100 |

Table B.5 reports the performance of LLM-only reward shaping across a variety of goal types. The model achieves near-perfect accuracy, precision, and recall, indicating a strong ability to generate consistent goal labels from environment states. The slight drop in precision for the *Go to a green door* task highlights occasional LLM hallucinations, though such cases are rare. Consequently, most residual inaccuracies in the overall reward shaping pipeline come from lightweight downstream models used to reduce computational overhead, rather than from the LLM-generated annotations themselves.

### B.6. Benchmark results per goal category

*Table B.6. Online evaluation of generalization performance split per goal category.* This is the **success rate [%]** presented in Table 1 with the 400 $(g, s_0^g)$ grouped by goal category.

| Method | Environment | Goals | Pick up a X | Go to the X | Open a X | Put an X next to a Y |
|---|---|---|---|---|---|---|
| Llama-3-8B (vanilla) | train/test | train/test | 11 (± 1.6) | 35 (± 2.3) | 8 (± 1.1) | 0 (± 0.0) |
| Llama-3-70B (vanilla) | train/test | train/test | 13 (± 1.7) | 33 (± 2.2) | 2 (± 0.5) | 0 (± 0.0) |
| Llama-3-8B (in-context+CoT) | train/test | train/test | 6 (± 1.3) | 36 (± 2.1) | 10 (± 1.4) | 0 (± 0.0) |
| Llama-3-70B (in-context+CoT) | train/test | train/test | 9 (± 1.4) | 45 (± 1.7) | 14 (± 1.2) | 0 (± 0.0) |
| TEDUO: steps 1 & 2 + BabyAI-IL-bot | train | train | 46 (± 2.0) | 92 (± 1.2) | 100 (± 0.0) | 7 (± 2.9) |
| | test | train | 30 (± 1.6) | 58 (± 1.7) | 100 (± 0.0) | 6 (± 2.1) |
| | train | test | 5 (± 1.2) | 44 (± 2.5) | 4 (± 0.9) | 0 (± 0.0) |
| | test | test | 7 (± 1.2) | 40 (± 2.2) | 5 (± 0.9) | 0 (± 0.0) |
| **TEDUO** (Llama-3-8B) | train | train | 46 (± 2.3) | 85 (± 1.3) | 100 (± 0.0) | 0 (± 0.0) |
| | test | train | 39 (± 2.4) | 65 (± 2.0) | 100 (± 0.0) | 0 (± 0.0) |
| | train | test | 20 (± 3.8) | 87 (± 3.2) | 83 (± 1.8) | 0 (± 0.0) |
| | test | test | 26 (± 3.0) | 70 (± 2.8) | 61 (± 2.6) | 0 (± 0.0) |

## B.7. Generalization to Grids of Varying Sizes

In this experiment, we fine-tune `Llama-3-8B-Instruct` using the TEDUO pipeline, leveraging training data collected from environments with grid sizes 1, 4, 6, and 9. Here, the grid size refers to the number of rooms it contains. Figure B.3 illustrates an example of a 5×5 grid (size 25). We evaluate the resulting model on larger environments with grid sizes 16 (4×4), 25 (5×5), and 49 (7×7), as reported in Table B.7.

Classical CNN-based RL agents such as `BabyAI-IL-bot` fail to generalize to larger grids due to the fixed size of their convolutional layers, which constrains them to a fixed grid size. In contrast, the vanilla LLM exhibits consistent performance across grid sizes, albeit at a lower level. The model fine-tuned with TEDUO achieves substantially improved success rates on grid sizes close to those seen during training. This demonstrates TEDUO's ability to facilitate generalization to more complex environments. However, as the grid size increases beyond the training distribution, the performance gradually declines, ultimately approaching that of the vanilla LLM on 7×7 grids.

Table B.7. *Online evaluation of generalization to larger grid sizes.* Results are averaged over 200 $(g, s_0^g)$ pairs.

| Method | Environment | Goals | Success Rate | Episode Length | Invalid Actions |
|---|---|---|---|---|---|
| | 4x4 | train/test | 13 (±0.8) | 905 (±6.2) | 33 (±0.2) |
| Llama-3-8B (vanilla) | 5x5 | train/test | 16 (±0.8) | 901 (±6.3) | 35 (±0.2) |
| | 7x7 | train/test | 19 (±0.8) | 874 (±6.8) | 39 (±0.1) |
| TEDUO: steps 1 2 + BabyAI-IL-bot | 4x4 / 5x5 / 7x7 | train/test | 0 (±0) | 500 (±0) | NA |
| | 4x4 | train | 53 (±1.8) | 597 (±16.1) | 33 (±0.6) |
| | 4x4 | test | 36 (±1.6) | 726 (±14.3) | 38 (±0.4) |
| **TEDUO**-Llama-3-8B | 5x5 | train | 35 (±1.8) | 739 (±16.3) | 36 (±0.6) |
| | 5x5 | test | 31 (±1.6) | 757 (±13.3) | 36 (±0.4) |
| | 7x7 | train | 19 (±1.3) | 867 (±10.2) | 31 (±0.4) |
| | 7x7 | test | 25 (±1.3) | 852 (±10.1) | 35 (±0.4) |

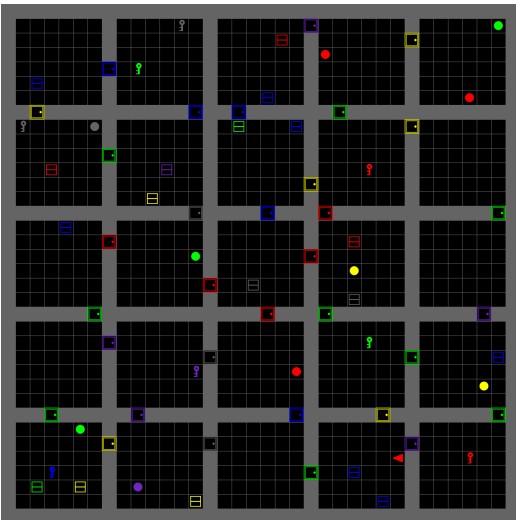

*Figure B.3. Example of a BabyAI grid of size 25 (5x5).*

# C. Experimental details

## C.1. Choice of Environments

**BabyAI.** This paper primarily uses the Minigrid-BabyAI environment to benchmark its method. This choice was motivated by several factors. Most importantly, we require a sandbox environment in which a wide range of goal reaching tasks can be expressed in natural language. Robotic environments ((Todorov et al., 2012; James et al., 2019)) were excluded due to precise control of robotic components being beyond LLM's prior knowledge and the need to discretize continuous actions to match LLM's tokenized output. Additionally, 3D environments ((Fan et al., 2022; Puig et al., 2018)) were not considered due to computational constraints. Text-based games ((Côté et al., 2018; Shridhar et al., 2021b)) were also excluded as they involve high-level text interactions, contrary to this paper's focus on low-level control task for language models.

Given the significant computational resources and time required to perform all three steps of our pipeline, in particular fine-tuning an LLM agent, our current scope is necessarily limited to BabyAI. Nonetheless, the insights derived from this controlled setting are broadly applicable and provide a foundation for future work in environments with similar tabular structures, such as NetHack ((Küttler et al., 2020)) and Overcooked ((Carroll et al., 2020)), which differ mainly in thematic focus (video game dungeon crawling and collaborative cooking, respectively).

**Webshop.** To demonstrate the ability of TEDUO to scale beyond grid-world environments, this section proposes an additional benchmark of TEDUO in the Webshop environment (Yao et al., 2023a). The development of digital agents which are can excel at tasks performed on computers, such as web navigation, coding, and operating office software is one of the most promising applications emerging from the LLM-Reinforcement Learning synergy. Webshop is a simulated e-commerce environment where an agent given product requirements must locate corresponding items by navigating a website. This environment is particularly relevant due to its dynamically evolving action space, presenting two key challenges:

**Dynamic UI Actions:** The available actions vary by state due to changes in the website's UI elements. We address this challenge by concatenating the available actions with the state description. **Linguistic Action Requirements:** The agent must generate linguistic inputs (e.g., search queries) to use the search bar. This highlights the necessity for the fine-tuned LLM to avoid catastrophic forgetting (Luo et al., 2025) and produce coherent keywords for narrowing searches.

## C.2. Guide for Practitioners

This section provides a high-level summary of the adaptations required to deploy TEDUO in a new environment. The key modifications involve adjusting the stages where domain knowledge is incorporated. Specifically, the following prompts must be adapted:

1. The prompt for generating the abstraction function (Appendix C.4).

2. The prompt for reward labeling (Appendix C.5).

3. The prompt used at inference time (Appendix C.7).

At these stages, prior knowledge about the environment's dynamics, the semantics of instructions, or any contextual information that can facilitate goal-conditioned policy learning in Step 3 can be integrated via natural language. The provided prompt examples illustrate how such adaptations can be implemented for two different environments.

Additionally, the offline Reinforcement Learning method used to solve the abstract Markov Decision Processes in Step 2 should be tailored to the characteristics of the environment. This includes considering factors such as modality, the scale of the state and action spaces, and the number of initial states.

## C.3. Data Collection

As mentioned in the main body of this paper, our pipeline makes no assumption on the data collection policy used.

### C.3.1. BABYAI ADAPTATION

In the experiments with the BabyAI environment, we rely on the default goal-oriented policies from the BabyAI environment. We denote these policies by $\pi^\beta(\cdot; g)$. Our data collection policy is then a random mixture of the policies $\pi^\beta(\cdot; g)$. Given a

randomly sampled initial state $x_0 \in \mathcal{X}$ and an unknown goal $g$ randomly sampled from the set of original BabyAI language commands, we let the agent interact with the environment according to $\pi^\beta(\cdot; g)$ until either $g$ is reached or the limit of 500 steps is reached. This policy simulates agents attempting to accomplish multiple task within the environment. Examples of real-world unlabeled data that could be generated from such policy include CCTV footage of employees at work, logs of medical procedures performed on a patient, or YouTube videos.

Refer to Appendix B.2 for an analysis of how the data collection policy affects our pipeline, comparing goal-oriented data collection with fully random data collection.

### C.3.2. WEBSHOP ADAPTATION

In the experiment with the Webshop environment, to demonstrate that our pipeline can learn from non-expert demonstrations, we design a goal-conditioned data collection policy, $\pi^\beta(\cdot; g)$, which relies on a pre-trained LLM and random actions. When encountering a "search" webpage that allows keyword input for locating the target object, $\pi^\beta(\cdot; g)$ prompts a pre-trained LLM with instruction $g$ to generate relevant keywords. In all other states, $\pi^\beta(\cdot; g)$ selects a random action from the set of available options provided by the environment. The pre-trained LLM is Llama-3-8B-Instruct with the following parameters: temperature: 1, top k: 40, maximum number of tokens: 200.

Each episode terminates either upon purchasing an object or reaching the step limit of 200. Using $\pi^\beta(\cdot; g)$, we collect 5,000 unlabeled trajectories spanning 1,500 unique instructions.

### C.4. Step 1: Abstraction Function

The abstraction function utilizes contextual understanding of LLMs to identify goal-relevant features. The state abstraction operator is implemented as a collection of Python functions built on top of the feature selection made by a prompted language model, $F(\cdot; g) = \text{LLM}^{abstrct}(g)(\cdot)$. Using LLM powered Python functions instead of directly applying the LLM to create an abstraction of each state reduces the number of LLM calls from $|\mathcal{X}||G^{tr}|$ to $|G^{tr}|$ and ensures that the abstraction is consistent across all states. The prompt for generating the code includes contextual information about the environment, a list of features, and a description of the goal, instructing the LLM to create a function that removes features of a state that are irrelevant to achieving the specified goal. Additionally, our state abstraction functions separate the set of relevant features into two subsets: $s_\phi^g$ and $s_{\bar{\phi}}^g$, so that $s_\phi^g \cup s_{\bar{\phi}}^g$. The features in $s_\phi^g$ are the ones which are necessary for identifying if the underlying low-level state achieves $g$, i.e. if $x \in \phi(g)$. The remaining features, which are relevant for solving the task specified by $g$ but not strictly necessary for identifying if this goal is achieved are found in $s_{\bar{\phi}}^g$. We introduce this separation of abstracted features to even further reduce the dimensionality of the state space for hindsight labeling (see next section).

### C.4.1. BABYAI ADAPTATION

To adopt the state abstraction function in for the BabyAI environment, we prompt an LLM with the given goal, a randomly sampled state representation as an example, and two in-context examples of the expected output. Figure C.4 shows the prompt template used. The LLM returns the goal relevant features which are then passed to a python function that processes states according to the following rules:

- Distractors identified in the selected features are labeled as either "goal object" or "goal location."

- Distractors not included in the selected features are labeled as obstacles.

- Doors not referenced in the selected features are assigned uniform colors.

- If all relevant objects are within the agent's current room, the environment outside the room is disregarded.

In our experiments, we use the *Llama-3-70B-Instruct* language model with the following parameters: {temperature: 0, top k: 1, maximum number of tokens: 8000}.

### C.4.2. WEBSHOP ADAPTATION

The WebShop environment includes predefined state abstractions that simplify the original HTML representation of the website. In our experiments, we utilize these abstractions, which are shared across all goals rather than being goal-specific,

as in BabyAI. Furthermore, given the demonstrated ability of LLMs to parse HTML (Gur et al., 2023), we are confident that TEDUO Step 1 could replicate a similar abstraction. Since WebShop is non-Markovian, we ensure Markovian properties by using historical states–concatenating all states from the initial to the current state. References to "states" in the following discussion refer to these history-augmented states.

```
<|begin_of_text|><|start_header_id|>system<|end_header_id|>You are helping a Reinforcement learning agent in the minigrid
    environment. Always answer as helpfully as possible, while being truthful.<|eot_id|><|start_header_id|>user<|
    end_header_id|>Given a grid, its features and a goal, can you simplify the features of the grid by detecting all the
    objects related to the goal and if necessary goal location. if necessary, make sure to flag all the relevant object and
    not just one.

I'm giving you two examples on the same grid:

Grid : "It is a 22 by 22 tiles grid. The features of the environment are:
0. The following tiles are wall: (1,7) (1,14) (2,7) (2,14) (3,7) (3,14) (4,7) (5,7) (5,14) (6,14) (7,1) (7,2) (7,3) (7,4) (7,5)
    (7,6) (7,7) (7,8) (7,9) (7,10) (7,11) (7,13) (7,14) (7,15) (7,16) (7,17) (7,18) (7,19) (7,20) (8,7) (8,14) (9,14) (10,7)
    (10,14) (11,7) (11,14) (12,7) (13,7) (13,14) (14,1) (14,2) (14,3) (14,4) (14,5) (14,6) (14,7) (14,9) (14,10) (14,11)
    (14,12) (14,13) (14,14) (14,16) (14,17) (14,18) (14,19) (14,20) (15,7) (15,14) (16,7) (16,14) (17,7) (17,14) (18,7)
    (18,14) (19,14) (20,7) (20,14)
1. A open purple box is on tile (1,20)
2. A open green box is on tile (5,8)
3. A open yellow box is on tile (6,5)
4. A open blue box is on tile (8,13)
5. A open purple box is on tile (15,3)
6. A open grey box is on tile (18,10)
7. A open red box is on tile (20,19)
8. A closed yellow door is on tile (4,14)
9. A closed purple door is on tile (6,7)
10. A locked grey door is on tile (7,12)
11. A closed red door is on tile (9,7)
12. A closed yellow door is on tile (12,14)
13. A closed grey door is on tile (14,8)
14. A closed grey door is on tile (14,15)
15. A closed red door is on tile (19,7)
16. A blue key is on tile (3,5)
17. A grey key is on tile (8,10)
18. A blue key is on tile (11,4)
19. A purple ball is on tile (1,16)
20. A green ball is on tile (2,20)
21. A blue ball is on tile (3,19)
22. A red ball is on tile (9,12)
23. A grey ball is on tile (9,13)
24. A yellow ball is on tile (13,1)
25. A grey ball is on tile (13,6)
26. A yellow ball is on tile (17,6)
27. Inventory : []

Exemple 1 :
The goal is "Pick up a blue key".

Following the indications, the correct output is these simplified features :

{"goal object" : (3,5) (11,4)}

Example 2 :
The goal is "Put a green box next to a grey ball".

Following the indications, the correct output is these simplified features :

{"goal object" : (18,10),
"goal location" : (9,13) (13,6),}

Now, my goal is "{goal}" and I am in the following grid :
"It is a 22 by 22 tiles grid. The features of the environment are:
{state}

Let's think step by step. First, tell me about your knowledge of the Minigrid/BabyAI reinforcement learning environment. Then,
    provide an analysis of the environment and the goal. Finally, write simplified features in the same format as the example
    .<|eot_id|><|start_header_id|>assistant<|end_header_id|>
```

*Figure C.4.* Prompt template for selecting the relevant features to achieve the goal in the BabyAI environment.

## C.5. Step 1: Reward Shaping

As detailed in Section 3.1.2, the reward shaping process involves two stages.

In the first stage, a large language model LLM, here *Llama-3-70B-Instruct*, is utilized to generate a supervised dataset of labeled goals, $\{(s^g, r^g) : r^g = \text{LLM}^{rwrd}(s^g; g), s^g \in \tilde{\mathcal{S}}^g\}$ for each $g \in \mathcal{G}^{tr}$. We choose $\tilde{\mathcal{S}}^g$ as a small (up to 5000 states), diverse subset of the abstract space $\mathcal{S}^g$ and $r^g \in \{0, 1\}$. The subset $\tilde{\mathcal{S}}^g$ is chosen so that for any two abstract states, the

```
0. The following tiles are wall: (1,7) (1,14) (2,7) (2,14) (3,7) (3,14) (4,14) (5,7) (6,7) (6,14) (7,1) (7,2) (7,3) (7,4) (7,5) (7,6)
      (7,7) (7,9) (7,10) (7,11) (7,12) (7,13) (7,14) (7,15) (7,16) (7,17) (7,18) (7,19) (7,20) (8,7) (8,14) (9,7) (10,7) (10,14) (11,7)
      (11,14) (12,7) (12,14) (13,14) (14,1) (14,2) (14,3) (14,4) (14,6) (14,7) (14,8) (14,9) (14,10) (14,11) (14,12) (14,14) (14,15)
      (14,16) (14,18) (14,19) (14,20) (15,14) (16,7) (16,14) (17,7) (17,14) (18,7) (18,14) (19,7) (19,14) (20,7) (20,14)
1. A open red box is on tile (1,2)
2. A open yellow box is on tile (4,9)
3. A open blue box is on tile (6,8)
4. A open grey box is on tile (16,15)
5. A open grey box is on tile (17,1)
6. A open red box is on tile (20,6)
7. A closed blue door is on tile (4,7)
8. A closed red door is on tile (5,14)
9. A closed purple door is on tile (7,8)
10. A closed blue door is on tile (9,14)
11. A closed yellow door is on tile (13,7)
12. A closed yellow door is on tile (14,5)
13. A closed red door is on tile (14,13)
14. A closed red door is on tile (14,17)
15. A closed grey door is on tile (15,7)
16. A grey key is on tile (5,20)
17. A yellow key is on tile (9,15)
18. A green key is on tile (15,5)
19. A yellow key is on tile (16,12)
20. A green key is on tile (17,15)
21. A red ball is on tile (4,19)
22. A purple ball is on tile (9,5)
23. A purple ball is on tile (12,2)
24. A blue ball is on tile (16,19)
25. Inventory : []
26. The agent is currently at the following tile: (6,10)
27. The agent is facing up
```

*Figure C.5. An example of BabyAI textualized state before state abstraction.*

```
The following tiles are wall: (1,7) (1,14) (2,7) (2,14) (3,7) (3,14) (4,14) (5,7) (6,7) (6,14) (7,1) (7,2) (7,3) (7,4) (7,5) (7,6)
      (7,7) (7,9) (7,10) (7,11) (7,12) (7,13) (7,14) (7,15) (7,16) (7,17) (7,18) (7,19) (7,20) (8,7) (8,14) (9,7) (10,7) (10,14) (11,7)
      (11,14) (12,7) (12,14) (13,14) (14,1) (14,2) (14,3) (14,4) (14,6) (14,7) (14,8) (14,9) (14,10) (14,11) (14,12) (14,14) (14,15)
      (14,16) (14,18) (14,19) (14,20) (15,14) (16,7) (16,14) (17,7) (17,14) (18,7) (18,14) (19,7) (19,14) (20,7) (20,14).
The following tiles are obstacles : (1,2) (4,9) (16,15) (17,1) (20,6) (5,20) (9,15) (15,5) (16,12) (17,15).
The following tiles are closed doors : (6,8) (4,7) (5,14) (6,8) (4,7) (5,14) (7,8) (9,14) (13,7) (14,5) (14,13) (14,17) (15,7).
A goal object is on the tile (4,19).
A goal object is on the tile (9,5).
A goal object is on the tile (12,2).
A goal object is on the tile (16,19).
Inventory : [].
The agent is currently at the tile (6,10).
The agent is facing up.
```

*Figure C.6. Textualized state from C.5 after applying state abstraction for the goal "pick up a ball".*

set of features relevant for goal-identifications is distinct, i.e. $\forall s_1^g \neq s_2^g \in \tilde{\mathcal{S}}^g, s_{1,\phi}^g \neq s_{2,\phi}^g$. This maximizes the chances of including goal-states in $\tilde{\mathcal{S}}^g$, mitigating the potential issue of generating a highly-imbalanced dataset for training our proxy neural networks.

In the second stage, a collection of neural networks $R_\theta(\,\cdot\,;g) : \mathcal{S}^g \rightarrow \{0,1\}$ indexed by $g \in \mathcal{G}^{tr}$ is trained on the constructed supervised datasets.

### C.5.1. BABYAI ADAPTATION

In our experiments with the Baby AI environment, the state-labeling LLM is configured with parameters {temperature: 0, top-k: 1, max tokens: 8000}, and we use the prompt template shown in Figure C.7.

The state representations are transformed from text to a grid format. The network architecture consists of a small convolutional neural network with one convolutional layer (output dimension: 32, kernel size: (2,2)), followed by two linear layers (hidden dimension: 32, output dimension: 1). A Sigmoid activation function is applied after the final linear layer, and ReLU is used after all other layers. Dropout layers are added before each linear layer. The network is trained with the following hyperparameters: learning rate of 1e-5, maximum of 3000 epochs, and dropout rate of 0.1. The dataset is split into training and validation sets (90%/10%), and the model weights with the lowest validation loss are retained.

```
<|begin_of_text|><|start_header_id|>system<|end_header_id|>You are a helpful and honest judge of good progress in the Minigrid/
    BabyAI reinforcement learning environment with respect to a specific GOAL. Always answer as helpfully as possible, while
    being truthful, simple and concise. If you don't know the answer to a question, don't share false information.
<|eot_id|><|start_header_id|>user<|end_header_id|>I will present you a GOAL to be achieved and the descriptions of a STATE of
    the environment. Examples of goal are "opening a door", "go to a specific location", "putting an object next to another
    other" or "picking up an object".
First, tell me about your knowledge of the Minigrid/BabyAI reinforcement learning environment related to the goal.
Then, write an analysis describing the semantics of the state strictly using information from the description and your
    knowledge of Minigrid/BabyAI.
Finally, respond by explicitly declaring if the state indicates that the GOAL has been achieved at any point in the past,
    writing either ("goal achieved": True), or ("goal achieved": False). If you have a doubt, you could also say ("goal
    achieved": NA).

The environment is a 22 by 22 tiles grid. An object that has been picked up is placed in the agent inventory.

The agent or an object is considered at an object location if it is on an adjacent tile to the object (for example, (4,2) and
    (5,3) are not adjacent as their Manhattan distance |4−5| + |2−3| = 2 is strictly superior to 1) or it is in the inventory
    . If the goal explicitly mentions the agent going to an object or putting an object near another object, compute the
    Manhattan distance, show the details of the computation, explicitly compare the result to 1 and then verify your
    reasoning does not have any mistakes and base your decision only on the Manhattan distance. Don't say they are adjacent
    if their Manhattan distance is higher than 1. Don't forget to check the inventory. If the coordinates of the destination
    are mentioned, the agent must go to this exact tile.

For other types of goals, do not compute them and ignore the previous paragraph.

{"STATE": {state}}

{"GOAL": {goal}}<|eot_id|><|start_header_id|>assistant<|end_header_id|>
```

*Figure C.7.* Prompt template for labeling states as goal states or not for the BabyAI environment.

### C.5.2. WEBSHOP ADAPTATION

In TEDUO Step 1, goal-conditioned reward labeling is performed by prompting an LLM to evaluate the alignment between the instruction and the purchased product based on four criteria: category, attributes, options, and price. These criteria directly influence the true reward function defined by the environment. The resulting scores are combined using the formula specified in the WebShop environment to generate the synthetic reward. The prompt includes five example instruction evaluations (see Figure C.8). We use Llama-3.1-70B-Instruct with the following parameters: {temperature: 0, top-k: 1, max tokens: 100}. Since only terminal states (i.e., post-purchase states) require labeling in WebShop, the computational cost of this step is significantly reduced.

```
<|begin_of_text|><|start_header_id|>system<|end_header_id|>You are a helpful and honest judge of the fit between an instruction
        for purchasing an object and the proposed object. Your evaluation is based on whether the object meets four criteria:
        category, attributes, options, and price. Follow these steps:

1. Identify Attributes and Options

- List the attributes mentioned in the instruction.
- List the options selected in the instruction.

2. Evaluate Each Criterion

- Category: Assign a value of 1 if the proposed object's product category matches the instruction; otherwise, assign 0.
- Attributes: Count the number of attributes from the instruction that are correctly matched by the proposed object.
- Options: Count the number of options selected by the user that are correctly matched by the proposed object.
- Price: Assign a value of 1 if the proposed object's price is lower than or equal to the price specified in the instruction;
        otherwise, assign 0.
<|eot_id|><|start_header_id|>user<|end_header_id|>{Example Instruction 1}}<|eot_id|>
<|start_header_id|>assistant<|end_header_id|>{Example criteria 1}}<|eot_id|>
<|start_header_id|>user<|end_header_id|>{Example Instruction 2}}<|eot_id|>
<|start_header_id|>assistant<|end_header_id|>{Example critera 2}}<|eot_id|>
<|start_header_id|>user<|end_header_id|>{Example Instruction 3}}<|eot_id|>
<|start_header_id|>assistant<|end_header_id|>{Example criteria 3}}<|eot_id|>
<|start_header_id|>user<|end_header_id|>{Example Instruction 4}}<|eot_id|>
<|start_header_id|>assistant<|end_header_id|>{Example criteria 4}}<|eot_id|>
<|start_header_id|>user<|end_header_id|>{Example Instruction 5}}<|eot_id|>
<|start_header_id|>assistant<|end_header_id|>{Example criteria 5}}<|eot_id|>
<|start_header_id|>user<|end_header_id|>{GOAL}}<|eot_id|>
<|start_header_id|>assistant<|end_header_id|>
```

*Figure C.8.* Prompt template for synthetic reward in the Webshop environment.

## C.6. Step 2: Offline Reinforcement Learning

### C.6.1. BABYAI ADAPTATION

In TEDUO's step 2, the abstract MDPs are solved using tabular Q-learning. For each goal $g$, a Q-value table $Q^g$ of size $|S^g| \times |\mathcal{A}|$ is constructed. The Q-values are updated iteratively using the Bellman equation:

$$Q^g_{new}[s_t, a_t] \leftarrow (1 - \alpha)Q^g[s_t, a_t] + \alpha \left( r^g[s_t, a_t] + \gamma \max_a Q^g[s_{t+1}, a] \right).$$

In our experiments for each environment, the learning rate $\alpha$ is set to 0.1, and the discount factor $\gamma$ is set to 0.7. Subsequent states $s_{t+1}$ are restricted to transitions observed in $\mathcal{D}$. Iterations stop when $||Q^g_{new} - Q^g||_\infty < \epsilon$, where $\epsilon = 1 \times 10^{-6}$.

### C.6.2. WEBSHOP ADAPTATION

In the WebShop variant, given the deterministic nature of the environment and its unique starting state, we employ an improved filtered Behavioral Cloning method to solve the abstract MDPs. For each training goal $g$, we identify the terminal state with the highest reward $r^g$ among the collected samples. The corresponding trajectory is then refined by removing any potential loops. Finally, trajectories with a synthetic reward below a predefined threshold (0.6) are discarded.

## C.7. Step 3: LLM Fine-tuning

TEDUO's step 3 involves fine-tuning a large language model using the generated supervised dataset $\mathcal{D}^{SFT}$. In this paper, the fine-tuned model is *Llama-3-8B-Instruct*. We use Low-Rank Adaptation ((Hu et al., 2021)) to reduce the compute cost. The hyperparameters used for the fine-tuning step are detailed in Table C.8. The model weights with the lowest validation loss are retained. The fine-tunings have been realized on a cluster of 4 A100 (80GB VRAM). The computing power provided in figure 5 is determined by multiplying the number of GPU hours by the peak Tflops (312 for A100 in bf16) and the estimated utilization rate (90%).

*Table C.8.* **Fine-tuning hyperparameters**

| Hyperparameter | Value BabyAI | Value Webshop |
|---|---|---|
| Batch size (per device) | 10 | 4 |
| Learning rate | 2e-5 | 2e-6 |
| Maximum Gradient norm | 0.3 | 0.3 |
| Warmup ratio | 0.01 | 0.01 |
| Maximum number of epochs | 3 | 20 |
| LORA rank | 512 | 512 |
| LORA alpha | 512 | 512 |
| LORA dropout | 0.1 | 0.1 |
| Split train/val ratio | 0.1 | 0.1 |
| Tensor type | bf16 | bf16 |

```
<|begin_of_text|><|start_header_id|>system<|end_header_id|>You are a Reinforcement learning agent in the minigrid environment.
     You select the sequence of optimal actions to achieve the GOAL. Always answer as helpfully as possible, while being
     truthful.<|eot_id|><|start_header_id|>user<|end_header_id|>The state of the environment is given by the STATE. The
     environment is a 22 by 22 tiles grid. The possible actions are { 0: turn left, 1: turn right, 2: move forward in the
     direction faced by the agent, 3: pick up an object, 4: drop an object, 5: toggle/activate an object, 6: done completing
     the task }.
You only output the list of numbers associated with the optimal sequence of action to achieve the GOAL.

STATE : {state}

GOAL : {goal}.<|eot_id|><|start_header_id|>assistant<|end_header_id|>
```

*Figure C.9.* This prompt template is employed to generate a sequence of optimal actions to achieve the given goal while being in the given state for the BabyAI environment.

## C.8. Evaluation Setup

### C.8.1. BABYAI ADAPTATION

**Environments.** An environment in this context refers to a grid setup, which includes the arrangement of rooms, doors, and objects. The training environments consist of the grid setups included $\mathcal{D}$. This implementation uses 40 distinct environments for training the model. Testing environments are entirely new grid setups not encountered during training. For this benchmark, we utilize 2 different grid setups for testing.

**Goals.** Training goals are defined as the goal contained in $\mathcal{G}^{tr}$, a subset of natural language instructions provided by BabyAI without any modifications. Testing goals differ both grammatically **and** semantically from training goals. They are derived from BabyAI's original instructions, distinct from $\mathcal{G}^{tr}$, and reformulated using alternative phrasings and synonyms. Tables C.9, C.10, and C.11 provide the alternative formulations and synonyms for objects and colors used in these reformulations.

**Baselines:**

**LLMs (vanilla).** The vanilla Large Language Model baseline utilizes *Llama-3-8B-Instruct* or *Llama-3-70B-Instruct* prompted with the template shown in Figure C.9. This prompt provides basic information about the environment, current goal, and a textual (non abstracted) representation of the state.

**LLMs (in-context + CoT).** This baseline extends the vanilla LLM approach by using the prompt in Figure C.10, which includes detailed environment information, similar to a game manual, as described in (Wu et al., 2023). It also integrates expert demonstrations using textual grid examples and goals with their optimal action sequences. The CoT prompting technique is employed to guide the LLM through multi-step reasoning and self-reflection.

**BabyAI-IL-bot.** This baseline employs the official implementation from (Chevalier-Boisvert et al., 2018) using Imitation Learning (IL) with the largest default model parameters: memory dimension = 2028, recurrence = 80, batch size = 768, instruction architecture = AttentionGRU, instruction dimension = 256, learning rate = $5 \times 10^{-5}$. Training is performed on the supervised dataset $\mathcal{D}^{SFT}$ from TEDUO step 2 instead of an expert demonstration dataset.

```
The state of the environment is given by the STATE. The environment is a {env[0]} by {env[1]} tiles grid. The possible actions
    are { 0: turn left, 1: turn right, 2: move forward in the direction faced by the agent, 3: pick up an object, 4: drop an
    object, 5: toggle/activate an object, 6: done completing the task}. An object that has been picked up is placed in the
    agent inventory. The agent or an object is considered at an object location if it is on an adjacent tile to the object (
    For example, (4,2) and (5,3) are not adjacent as their Manhattan distance |4−5| + |2−3| = 2 is strictly superior to 1) or
    it is in the inventory. If the coordinates of the destination are mentioned, the agent must go to this exact tile. Make
    sure you are facing the right direction before using the action "2".

You only output the list of numbers associated with the optimal sequence of action to achieve the GOAL.

To help you achieving the GOAL, I provide examples of optimal sequences of actions for multiple examples GOAL with different
    examples STATE.

###Example 1 :

GOAL : {Example goal 1}.

STATE : {Example state 1}.

Sequence of actions : {Example action 1}

###Example 2 :

GOAL : {Example goal 2}.

STATE : {Example state 2}.

Sequence of actions : {Example action 2}

Now, I will present you a GOAL to be achieved. First, tell me about your knowledge of the BabyAI reinforcement learning
    environment. Second, explain how you can use the proposed actions to move around the grid. Third, similar to the example,
    output a Python list that contains the sequence of action keys (1−6) chosen to achieve the goal.

GOAL : {goal}.

STATE : {state}.
```

*Figure C.10.* This prompt template is employed to generate a sequence of optimal actions to achieve the given goal while being in the given state. It uses in-context learning and CoT prompting for the BabyAI environment.

### C.8.2. WEBSHOP ADAPTATION

**Environment.** Since the WebShop environment does not provide multiple configurations, our evaluation focuses on generalization to new goals.

**Goals.** The training goals $\mathcal{G}^{tr}$ are a subset of the natural language instructions provided by the WebShop environment, used without modification. Similarly, the testing goals form a distinct subset of instructions generated using different seeds while ensuring no overlap with $\mathcal{G}^{tr}$. Given the grammatical and semantic diversity of the instructions, we do not perform any goal reformulation.

**Baselines:**

**ReAct.** This baseline leverages general-purpose LLMs with the ReAct prompting technique (Yao et al., 2023b), which is state-of-the-art for low-data settings in this environment. ReAct enhances reasoning in LLMs by integrating reasoning traces with action steps within the same prompt, enabling the model to reason through complex problems while simultaneously retrieving and validating relevant information. This feedback loop improves both accuracy and coherence. Vanilla prompting was excluded, as it failed to reach the purchasing step within the step limit (15), preventing it from achieving valid rewards.

*Table C.9. Alternative formulations for the natural language commands.*

| Original instruction | Alternative formulation |
| --- | --- |
| Go to the tile (X,Y) | Move to the location at the coordinate (X,Y) / Reach the position at (X,Y) / Navigate to the point (X,Y) |
| Pick up a X | Grab a X / Acquire a X / collect a X |
| Go to a X | Move to a X / Reach a X / Naviguate to a X |
| Open a X | Push a X open / Swing open a X |
| Put a X next to a Y | Set a X and a Y next to each other / Position a X alongside a Y / Place a X beside a Y |

*Table C.10. Synonyms used for the objects.*

| Original word | Synonyms |
|---|---|
| Box | Container / Crate / Chest |
| Key | Passcode / Lock-opener / Unlocker |
| Ball | Sphere / Globe / Orb |
| Door | Portal / Gate / Hatch |

*Table C.11. Synonyms used for the colors.*

| Original Color | Synonyms |
|---|---|
| Blue | Azure / Cobalt / Navy |
| Red | Scarlet / Crimson / Ruby |
| Green | Emerald / Jade / Lime |
| Yellow | Golden / Amber / Canary |
| Purple | Violet / Lavender / Mauve |
| Grey | Ash / Charcoal / Silver |

