# OpenReview forum: "The Synergy of LLMs & RL Unlocks Offline Learning of Generalizable Language-Conditioned Policies with Low-fidelity Data"
_ICML.cc/2025/Conference — ICML 2025 spotlightposter_

### Official Review · Reviewer_zuF6 · 2025-03-10

**Overall Recommendation:** 4

**Summary:**

Existing reinforcement learning (RL) approaches often struggle to generalize to unseen goals and states. To address it, this paper propose TEDUO,  a training pipeline for offline language-conditioned policy learning in symbolic environments.  TEDUO employs large language models (LLMs) as generalizable instruction-following agents. Experimental results demonstrate that TEDUO outperforms baselines.

## update after rebuttal
I have carefully read all the reviewers' comments as well as the authors' rebuttal. Most of my concerns have been addressed. So I rise the score to 4.

**Claims And Evidence:**

The main claims of this paper are that TEDUO can: 1) Ground LLMs for Multi-Step Decision Making, and 2) Enhance Generalization and Data Efficiency. These claims are supported by the experimental results.

**Essential References Not Discussed:**

The paper should introduce more RL-based policies and clearly differentiate them from the RL method used in TEDUO.

**Experimental Designs Or Analyses:**

As stated in the Methods and Evaluation Criteria section, the paper lacks sufficient RL-based policies as baselines, especially prior SOTA methods on BabyAI and Webshop.

**Methods And Evaluation Criteria:**

The proposed method is reasonable and expected to be effective. However, in the experimental design, comparisons are only made with the non-finetuned LLM backbone and certain baselines, lacking sufficient RL-based policies as baselines.

**Other Comments Or Suggestions:**

In Figure 1, the text describing the three key steps could be arranged horizontally to improve readability and help readers quickly understand the process.

**Other Strengths And Weaknesses:**

Strength

1. This paper is well-written, with good presentation and clear motivation, making it easy to follow.

2. The paper conducts sufficient ablation studies to support its claims and demonstrate the effectiveness of the proposed method.

3. Using RL-based policies to distill LLMs is an interesting approach and may provide insights for future LLM-based policy research.

Weakness

1. The paper conducts experiments on relatively simple environments such as BabyAI and Webshop, which are insufficient to demonstrate the proposed method's generalization to more challenging online environments like Habitat, RLBench, or Minecraft.

**Questions For Authors:**

1. In line 078, how are these goals obtained? Are they manually designed?

2. What is the temperature setting for the LLM? Would a high temperature setting affect the LLM's performance (either improving or degrading it)?

**Relation To Broader Scientific Literature:**

I did not find particularly relevant literature, aside from works on large language models and reinforcement learning policies.

**Theoretical Claims:**

The theoretical claims explained in Section 2 do not present any obvious issues.

---

> ### Author Rebuttal · Authors · 2025-04-01
>
> Thank you for your feedback on our work. Below, we would like to address your questions and concerns.
>
> ---
>
> ### P1 Additional baselines.
>
> We understand that a key concern in your review is the lack of additional baseline comparisons. However, to the best of our knowledge, no existing standard RL methods operate under the exact same setting as TEDUO beyond those already included in our benchmarking. Prior work typically requires either access to an online environment, something we explicitly rule out due to our focus on offline training paradigms or relies on labeled demonstration data, whereas we work with only an unlabeled set of state-action transitions.
>
> - **New result:** As suggested by reviewer `AzTQ`, we include an additional comparison with DeepSeek-R1, not available at the time of our original submission. We have incorporated its performance into our benchmark; results under this [link](https://imgur.com/a/wxLwFgh).
>
> If the reviewer is aware of any other works that can be applied in our setting, we would be eager to include them in our comparisons to further strengthen our evaluation. We would be grateful for any suggestions!
>
> ### P2 The complexity of tasks and environments.
>
> We appreciate the reviewer’s concern regarding the complexity of our evaluation environments. BabyAI and WebShop were deliberately chosen as two distinct yet illustrative environments: a grid-based world and a text-based web interaction task. This illustrates TEDUO’s adaptability across symbolic environments of different kinds. Additionally, as detailed in Appendix C.1, WebShop presents significant challenges, such as dynamic UI-based actions and linguistic action requirements, making it a non-trivial testbed.
>
> TEDUO's modularity makes it readily generalizable to other symbolic environments, as outlined in Appendix C.2 (*Guide for Practitioners*). While environments such as RLBench habitat and Minecraft are indeed complex, they are also image-based, which diverges from TEDUO’s focus on symbolic, text-based tasks. As acknowledged in our limitations section, continuous control tasks like RLBench may not be best suited for LLM-based agents. However, we recognize the potential for extending TEDUO’s applicability via, e.g., VLMs. We leave such extensions for future work.
>
> While scaling to more complex settings is an important goal, it is a common practice in RL research to focus on simple environments for a detailed and interpretable analysis. Notably, many works recently published at top-tier venues still focus on toy environments. For instance, this includes the relevant related work [1] cited by reviewer **azTQ,** which uses a card game as their primary environment for experimentation, and the related works cited by reviewer **vbjR** [1,3,4], which use environments much simpler than Webshop. Our focus on these settings allowed us to better understand TEDUO’s core mechanisms, providing a foundation for future extensions to more complex settings.
>
> Finally, while BabyAI and WebShop may appear toyish, we respectfully argue that the significance of our results goes beyond the complexity of these benchmarks. To the best of our knowledge, TEDUO is the first method capable of learning generalizable, language-conditioned policies from fully unlabeled data, which we find a significant and exciting result. The presented results demonstrate the promise of combining LLMs with RL-based approaches to enable flexible generalization.
>
> - **New Results:** To deepen the argument about flexible generalization, we have performed a new experiment demonstrating TEDUO’s generalization to *larger, unseen BabyAI grids* after training on smaller grids. Performance remains robust for grids three times larger than the grids seen during training. Results are available under this link : [link](https://imgur.com/a/kjgK47J).
>
> ### P4 How are the goals in $\mathcal{G}^{tr}$ obtained?
>
> The goals are manually designed (see  Appendix C.8.1 and C.8.2, paragraphs “Goals” for details).
>
> ### P5 What is the temperature setting for the LLM?
>
> We used the default temperature of the Llama-8B-Instruct model: 0.7 (both for TEDUO-fine-tuned and baselines).
>
> Motivated by your suggestion, we ran an additional small-scale experiment using online evaluation. Results, available at this [link](https://imgur.com/a/gxPcSbQ), show a slight improvement for temperature=1.2. We interpret this as a higher temperature promoting more diverse actions, helping the agent not to get stuck in suboptimal behaviors.
>
> Please note, tuning of the temperature is feasible when simulators are available. In offline RL scenarios, model selection is complicated, and hyperparameter tuning may not be possible.
>
> ---
>
> We sincerely appreciate the reviewer’s time and thoughtful feedback. We hope that our clarifications and new results address your concerns and strengthen the contributions of our work. Please let us know if there are any additional points that would benefit from further elaboration!

---

> > ### Comment · Reviewer_zuF6 · 2025-04-02
> >
> > Thank you for providing additional experimental results and clarifications. Most of my concerns have been addressed. If the remaining concern mentioned below can also be resolved, I will consider increasing my score.
> >
> > ## Q1: Additional baselines
> > The reviewer understands that identifying a suitable offline RL method for training with unlabeled data is a challenging task. However, if the first step of TEDUO is viewed as utilizing an LLM to label trajectories, then many existing offline RL methods could serve as reasonable baselines—assuming they are all trained on the data obtained from this initial step of TEDUO. Including more baseline comparisons could help make the contribution of this work more solid and convincing.

---

> > > ### Author Response · Authors · 2025-04-02
> > >
> > > Dear reviewer,
> > >
> > > We are glad that our previous response addressed most of your initial concerns.
> > > We agree that once the first step of TEDUO is completed, it is in principle possible to apply classical offline RL methods. However, our setting introduces two key complexities: (1) dealing with natural language goals, and (2) generalizing across a diverse set of tasks, as required in goal-conditioned reinforcement learning (GCRL). These aspects make standard offline RL methods less directly applicable. In particular, scaling value estimation in the presence of multiple goal-conditioned reward functions remains an open challenge.
> > >
> > > As an example, recent efforts to adapt Implicit Q-learning (IQL) to GCRL (excluding the natural language aspect) have succeeded only in maze navigation [1]. To our knowledge, IQL has not been successfully extended to natural language goal-conditioned RL, likely due to the scalability issues mentioned above. We think it is an exciting and valuable direction for future research.
> > >
> > > In our work, we searched extensively for applicable offline RL baselines for the BabyAI environment and found that the only viable option remains imitation learning using an LSTM+CNN policy, as used in our experiments. This is consistent with recent literature, where this approach is also the sole baseline adopted in comparable settings [2,3,4].
> > >
> > > If the reviewer is aware of other relevant offline RL baselines applicable to training natural-language goal-conditioned policies in BabyAI, we would be happy to include and evaluate them in a revised version. We hope this answers your last concern, and we are happy to discuss further if needed.
> > >
> > > [1] (2024). Navigation with QPHIL: Quantizing Planner for Hierarchical Implicit Q-Learning
> > >
> > > [2] (Neurips 2022). Pre-Trained Language Models for Interactive Decision-Making
> > >
> > > [3] (RL workshop ICLR 2022). Zero-Shot Compositional Policy Learning via Language Grounding
> > >
> > > [4] (RL workshop ICLR 2023). Unified Policy for Interleaving Language Reasoning with Actions

---

### Official Review · Reviewer_GwQb · 2025-03-12

**Overall Recommendation:** 4

**Summary:**

This paper introduces TEDUO, a method for fine-tuning instruction-following LLM agents using an unlabeled dataset of interactions (i.e. environment transitions without instructions or rewards).

TEDUO operates in two key stages:
1. The LLM labels the dataset by determining whether any possible goals are reached in each transition.
2. Tabular Offline Reinforcement Learning (RL) is then applied to learn one policy per goal. These policies generate a new expert demonstration dataset, which is used to fine-tune the LLM via imitation learning.

Empirical results on BabyAI with Llama-3-8B demonstrate strong generalization and impressive performance.

## update after rebuttal
As mentioned in my rebuttal comment, I deeply appreciate the effort put into this rebuttal, including the additional experiments and several updates to the manuscript. While I believe some empirical results—such as applying Imitation Learning directly to the demonstrations—might still be useful (even though the authors’ explanation for why this would lead to poor results is reasonable), the authors have addressed most of my concerns. I am therefore now recommending acceptance of this paper.

**Claims And Evidence:**

One important contribution of the paper is proposing an approach to leverage an unlabelled dataset of demonstrations. This approach leverages an LLM to label data given a known set of goals. First, several prior works proposed similar approaches, such as MineDojo (Fan et al., 2022) or LMA3 (Colas et al., 2023). Then, none of the experiments really leverages a true dataset of unlabelled demonstrations as demonstrations were collected using goal-conditioned policies. In comparison, MineDojo did use unlabelled recordings of humans. Finally, as the true labels are known, several natural questions arise e.g. What is the accuracy of the LLM as goal labeller? Did the goal labeller also found intermediate goals achieved? However, no analysis is provided on this key part of the method. Appendix B.5 indicates the performance of the lightweight neural networks trained to reproduce the LLM's labelling, but the accuracy seems to use the LLM's labels as ground truth.

The authors also argue that TEDUO is very robust to the quality of demonstrations. Appendix B.2 proposes an analysis of this, and I am really struggling to understand the results. In particular, the authors seem to say that collecting demonstrations with a random policy leads to better results than using an optimal policy for some goals. This appears surprising, and I do not understand how to relate the metric studied (the "number of unique initial states for which a goal is reachable") to TEDUO's performance. Additionally, even though these results seem to show that using an optimal goal-conditioned policy to collect demonstrations is not optimal for TEDUO, the authors still used it in the experiments in the main paper.

Finally, the authors argue that they study, in comparison to prior works, generalization on "semantically distinct" goals. While they do not define what they mean by "semantically distinct", Table B.5 hints that all types of goals were in the training set for their experiments on BabyAI, meaning that what differs between the training and test set is the name of the objects that agent must interact with (which is not different from what prior works did). Moreover, the authors argue that prior work mostly studied generalization on synonymous instructions, but they also introduce such goals by asking GPT-4 to paraphrase the original commands from BabyAI.

**Essential References Not Discussed:**

I do not see any essential references that were not discussed.

**Ethics Expertise Needed:**

["Responsible Research Practice (e.g., IRB, documentation, research ethics, participant consent)"]

**Experimental Designs Or Analyses:**

Results from Section 5.2 indicate that the policies learning with Offline RL perform poorly. Yet, using them to collect demonstrations for the LLM to perform Imitation Learning leads to impressive results. This seems quite surprising. One could say that, even though the trajectories generated by the policies are of poor quality, it is sufficient for the LLM to adapt its strategy successfully. But even more surprisingly, using these demonstrations to perform Imitation Learning with BabyAI-IL-bot also leads to good results (on the training environment/goals at least). Could the authors comment on this?

**Methods And Evaluation Criteria:**

The baselines chosen to compare TEDUO against, as well as the ablations with TEDUO + BabyAI-IL-bot (instead of the LLM) and the ones in Section 5.2, appear well chosen to me and very insightful.

I believe other key ablations would be very important. For instance, what would the performance of the LLM directly finetuned with Imitation Learning be on the demonstrations (which should be of much lower quality than the ones collected with the policies trained with Offline RL)? What would be the performance of directly applying Offline RL to the LLM based on the demonstrations instead of first learning intermediate policies? A method such as ILQL (Snell et al., 2023) could be used here.

Little is said about how the policies trained with Offline RL are used to produce the dataset for Imitation Learning. Section 3.3 seems to indicate that no interaction with the environment is performed and that an "empirical transition function" is used. More explanations on this part would be more than helpful.

I also have concerns regarding the experiments with WebShop. In particular, Appendix C.6.2 indicates that the Offline RL part of TEDUO was replaced by "filtered Behavioral Cloning." This raises a question about whether Offline RL is really necessary for TEDUO and also questions the choice of the WebShop if parts of the method are not suited for it.

**Other Comments Or Suggestions:**

The state abstraction part is said to be optional, yet it is the first subsection of the method section and abstract states are often referenced. I think this could be much clearer by either not mentioning abstract states too much in the main paper or always using abstract states (and not saying this part is optional) and putting the ablation on their usefulness in the main paper.

Their are a few minor typos I spotted:
    - l.223: "We require a controlled with environment"
    - l.881: "an import ant metric"

**Other Strengths And Weaknesses:**

I appreciated the authors performed multiple ablations and generalization experiments.

**Questions For Authors:**

I do not have any further questions.

**Relation To Broader Scientific Literature:**

As explained before, I believe the contribution of hindsight relabelling is overstated, given how close it is to prior works.
Apart from this, the literature is well-covered, including techniques similar but not applied in this paper, such as Inverse RL.

**Theoretical Claims:**

There are no theoretical claims.

---

> ### Author Rebuttal · Authors · 2025-04-01
>
> We thank the reviewer for their thoughtful feedback and valuable suggestions. Below, we address each of the key concerns and clarify aspects of our work.
>
> ---
>
> ### P1 Hindsight Labelling
> We acknowledge that LLM-based hindsight labeling itself is not novel, and we did cite prior work (Appendix A.2). The reviewer’s suggested references (MineDojo, LMA3) will also be added. However, these operate online, while TEDUO is fully offline. Our contribution also lies in scaling LLM-based reward functions by approximating them with NNs to reduce LLM calls. **Action:** We will refine our contribution statement to avoid overclaiming and emphasize these specific contributions.
>
> ### P2 LLM as a Goal Labeler
> We clarify that the results presented in B.5 report the accuracy of the rewards predicted by the NN reward approximators **with respect to the environment ground-truth rewards.** **Action:** We have additionally included the accuracy, precision, and recall metrics comparing LLM reward labeling to ground truth rewards. See this [link](https://imgur.com/a/Q2fixHr) for results.
>
> ### P3 Appendix B.2
> We acknowledge the reviewer's difficulty in interpreting Appdx. B.2, and we agree that the definition of $|\mathcal{S}_0^g|$ was unclear. **Action:** We have provided a more detailed definition and the motivation behind this metric. The revisions can be found under this link: [link](https://imgur.com/a/8pGgMIx). Thank you for drawing our attention to this part!
>
> ### P4 Experimental designs
> **Unlabeled Demonstrations** The claim that "none of the experiments leverage a true dataset of unlabeled demonstrations" is incorrect. We always act as if the data collection policy were completely unknown. Even if the policy used is a mixture of goal-conditioned policies, we do not assume knowledge of which trajectory has been generated with respect to which goal.
>
> **Semantically Distinct Goals** We define two "semantically distinct" instructions as two goals with different goal states, rather than paraphrased versions of the same task. (E.g., ”Pick up a red box” is distinct from “Pick up a yellow key” and is semantically equivalent to “Collect a red container”). While prior works have mostly focused on paraphrased instructions, we specifically test whether TEDUO can generalize to goals that differ in key task attributes. While this may seem incremental, we note that conventional RL approaches have struggled with this form of generalization. Please also consult P2 in the answer to reviewer `zuF6` for **new results** on generalization to more complex tasks.
>
> **Construction of $\mathcal{D}^{SFT}$.** We confirm that no additional interaction with the environment occurs in the construction of $\mathcal{D}^{SFT}$; an empirical transition function is used. This is due to our focus on purely offline RL.
>
> ### P5 Fine-tuning with Imitation Learning.
> See P5 in the answer to reviewer `vbJR`.
>
> ### P6: Directly applying Offline RL
> We appreciate the reviewer’s suggestion of applying offline RL directly to the LLM. While promising, this poses challenges due to scale and the natural language goal-conditioned setting.
>
> ILQL is an interesting adaptation of Implicit Q-Learning, but it targets standard RL tasks with a single reward function, not GCRL, which requires learning multiple reward functions and generalizing across tasks. Scaling value estimation in GCRL remains an open challenge.
>
> Recent efforts to adapt IQL to GCRL (excluding the natural language aspect) have succeeded only in maze navigation [1]. To our knowledge, ILQL has not been extended to natural language GCRL, likely due to scalability issues. We agree this is a valuable research direction.
>
> ### P7 The necessity of Offline RL
> We would like to highlight that filtered BC significantly improves performance (from an average score of 5 to 50 out of 100), demonstrating its effectiveness. A detailed discussion on the necessity of offline RL can be found in P5 in the answer to reviewer vbJR.
>
> ### P8 Performance of offline RL Policies vs. Final Results
> We would like to clarify that the average performance of the offline RL policies is low due to evaluation on *unseen initial states,* i.e. states that have not been visited in $\mathcal{D}$.  In such states, the Q-learning policies can only take random actions, leading to poor performance. For fine-tuning the LLM agent, we only use successful transitions to perform imitation learning via SFT. That is, $\mathcal{D}^{SFT}$ only includes trajectories that lead to goal achievement, which we can generate without access to online interaction with the environment by relying on the empirical state transition function and the learned reward functions.
>
> ### P9 State abstraction
> See P5 in the answer to reviewer `aztQ`.
>
> ---
>
> We thank the reviewer for their detailed and insightful feedback. We are confident that the revisions we plan to incorporate will further clarify and strengthen the paper and we are happy to answer any further questions you may have!

---

> > ### Comment · Reviewer_GwQb · 2025-04-02
> >
> > I thank the authors for their response.
> > I deeply appreciated the effort put into this rebuttal, including additional experiments and links to updated paragraphs.
> >
> > While I believe some empirical results such as using Imitation Learning directly on demonstrations might be useful (even though the authors' answer on why this would lead to poor results totally makes sense), the authors answered most of my concerns.
> > I am therefore increasing my score.

---

### Official Review · Reviewer_vbjR · 2025-03-13

**Overall Recommendation:** 4

**Summary:**

The paper studies the problem of training generalizable RL policies with the help of language models. RL policies can generally achieve impressive performance given enough exploration/coverage over the (state, action) space, but if the RL policy networks are trained from scratch on a particular environment, they can perform very poorly at new environments. On the other hand, pre-trained large language models show remarkable capabilities for generalization and are becoming more and more popular at decision making tasks.

This paper provides a recipe for training a language model on decision making tasks. Specifically, the paper uses LLMs to convert classical RL tasks into a text based representation and constructs solvable MDPs, next it uses tabular Q learning to solve these MDPs and learn optimal actions, and finally teaches an LLM these optimal actions for a given set of states and goals. The paper demonstrates that the resulting LLM learns generalizable strategies that can then transfer zero-shot to similar but unseen environments.

# Update after rebuttal

I like the paper's method and results --- and also the authors answered my concerns during the rebuttal process.

**I maintain my score at 4. I think this is a strong paper, and would strongly recommend its acceptance**.

**Claims And Evidence:**

Yes, the paper makes sound claims supported by detailed experiments, at least according to my opinion.

**Essential References Not Discussed:**

Nothing that I can think of.

**Experimental Designs Or Analyses:**

I checked the experimental design/analysis part of this paper, it made sense to me.

**Methods And Evaluation Criteria:**

Yes, the proposed methods and/or evaluation criteria makes sense.

**Other Comments Or Suggestions:**

None that I think of, please see my questions below and I would appreciate if the authors can address them!

**Other Strengths And Weaknesses:**

# Strengths

I think the most important strength of this paper is that they show generalization across new environments --- a key feature that traditional RL policies seem to lack. Pretrained LLMs have world knowledge and can learn strategies from a few environment but then are able to dynamically adapt to similar but new environments. These remarkable generalization potential can lead to general purpose decision making agents. Though the paper demonstrates this in a very toy setup, I think it is a powerful result and should be studied more by subsequent work.

# Weaknesess

1. Experiments are in very toy setup. It would be interesting to see if these results also hold in a more versatile set of environments.

2. (**Minor**) The paper title is very hard to understand and honestly, it seems to be LLM generated. Could the authors come up with a more suitable title?

**Questions For Authors:**

## Questions about base/instruction tuned models

1. Is the paper using the pretrained models (Llama-3-8B) or the instruction tuned versions (Llama-3-8B-Instruct)?

2. If the paper is using pretrained models, any particular reason why instruction tuned versions were not used? Could the authors give a comparison with the instruction tuned model as well?

3. It is possible that the base pretrained models are just bad at following instructions and hence we see a large performance improvement in Table 1. If one fine-tuned Llama-3-8B-Instruct instead using this paper’s method, how much improvement would one observe?

## Questions about Table 2

Do the authors have any insights on why the invalid actions for Llama-3-8B are double that of invalid actions? Can it be problematic in certain situations/can it be improved in some ways.


## Question about training on these tasks directly

I understand that for certain tasks, interacting with the environment or making a simulator is very difficult. But for the tasks that this paper experiments with, I imagine one could directly generate trajectories from the task and train the model on them. How would it perform compared to this paper’s complicated procedure/what benefit does this paper’s method have over that? For example, [1] (**concurrent work, so the authors need not cite it**) has similar ideas of using LLMs for general sequential decision making agents, but their pipeline seems significantly simpler compared to this paper’s method. Could the authors discuss this issue?

**Overall I am excited about this paper, and happy to recommend its acceptance pending my questions above are answered satisfactorily.**

# References

[1] Training a Generally Curious Agent, https://arxiv.org/abs/2502.17543

[2] ReAct: Synergizing Reasoning and Acting in Language Models, https://arxiv.org/abs/2210.03629

[3] Grounding Multimodal Large Language Models in Actions, https://arxiv.org/abs/2406.07904

[4] ELLA: Exploration through Learned Language Abstraction, https://arxiv.org/abs/2103.05825

**Relation To Broader Scientific Literature:**

# Language Conditioned RL

Early works established that language can specify goals for RL, but they often relied on expensive data gathering or manual labeling. For an example work on using language abstraction for decision making tasks (or the general flavor of work in this area), please see [4].

# LLM grounding in real tasks

With the rise of large language models, researchers began using off-the-shelf LLMs as decision-makers in interactive tasks. For example, Yao et al. (2023) proposed treating a general-purpose LLM as an agent that can choose actions by generating and evaluating plans (the ReAct framework) [2]​. Techniques like chain-of-thought prompting (CoT) and self-reflection have been shown to improve LLMs’ planning and reasoning on complex, multi-step tasks​. Despite these improvements, recent analyses (e.g. by Szot et al., 2024 [3]) found that prompting alone is insufficient for long-horizon decision-making in dynamic environments. One can use in-context learning or fine-tuning to achieve grounding in an LLM. Due to certain limitations, the authors of this paper move away from this direction, and propose the grounding via fine-tuning direction.

**Theoretical Claims:**

The paper has no theoretical claim of note to discuss.

---

> ### Author Rebuttal · Authors · 2025-04-01
>
> Thank you for your enthusiastic review and constructive feedback! We’re delighted by your positive assessment and have addressed your questions below to strengthen the manuscript further.
>
> ---
>
> ### P1 Task Complexity
>
> See P2 in answer to reviewer `zuF6`, showing **new results** on generalization from simple to more complex tasks.
>
> ### P2 Questions: Base vs. instruction-tuned models
>
> 1. We used **Llama-3-8B-Instruct** throughout all experiments (Appendix A.1), which we will clarify in the main text to avoid confusion.
> 2. Since we already use the instruction-tuned version, this question does not apply.
> 3. Similarly, since our experiments already use the instruct model, we think a comparison with the base model is not necessary.
>
> ### P3 Invalid action rate
>
> Pure RL policies (e.g., GCRL, tabular Q-learning policies) are designed to only take actions that have been seen in the training dataset (unless they are in a previously unseen state where the action is taken at random), minimizing invalidity. In contrast, in TEDUO, the LLM’s prior over the action space (which may contain invalid actions) is merged with the learned policies via SFT. While, on the one hand, this enables the LLM to flexibly generalize to new tasks and environments, it may occasionally lead to proposing invalid actions. As demonstrated in our experiments, the invalid action rate increases as the inference setting diverges from the training distribution; a non-fine-tuned LLM has a very high rate of invalid actions. This reflects a trade-off between generalization and memorization of the observed behaviors. When the agent is in a previously seen state, a potential workaround could integrate action masking by pruning options deemed invalid by the Q-learning policies. **Action:** We will discuss these observations in the limitations section of the revised manuscript, particularly in relation to high-stakes environments where safety is a concern.
>
> ### P4 Paper Title
>
> We appreciate the reviewer’s feedback regarding our title. While it was not LLM-generated (we promise!), we are open to suggestions for improving clarity in the camera-ready version.
>
> ### P5 Training directly on collected trajectories vs. TEDUO.
>
> We interpret the reviewer’s question as suggesting an alternative approach where the LLM is fine-tuned directly on observed trajectories. However, this approach has several limitations:
>
> Firstly, the original data collection policy used to generate the set of trajectories used for subsequent offline RL may be highly suboptimal with respect to the test time goals. In fact, it can even be random (see Appendix B.2. for an experiment with randomly collected observational data).
>
> Secondly, the generated trajectories a priori are not goal-labeled. While we could, in theory, fine fine-tune the LLM on sequences: $s_0, a_0, s_1, a_1, \ldots$, the LLM could not be made aware of which goal is the given sequence solving. This challenge is addressed by the hindsight labeling step of our pipeline.
>
> Finally, skipping the policy learning stage and fine-tuning the LLM on just the trajectories matched with the goals after step 1 is expected to perform strictly worse than fine-tuning the LLM on the Q-learning policies. This is a consequence of the suboptimal nature of the data collection policies. Importantly, this Q-learning step is also the most computationally inexpensive part of our pipeline, as it does not involve any LLM calls, so including it only marginally increases the overall compute.
>
> ## P6 Concurrent work
>
> Regarding the concurrent work cited by the reviewer [1], we find it an interesting idea, but we also highlight some key differences:
>
> - In the first part of this method, the collection policy (an LLM) is assumed to be good enough to generate a large number of trajectories with positive reward. (in our paper the step 2 (offline RL) goal is to obtain more positive trajectories based on the suboptimal offline data).
> - In the second part, they shift to online interaction, generating trajectories from their trained models, whereas TEDUO remains strictly offline throughout.
> - This method does not handle unlabelled observational data, while TEDUO explicitly addresses this by leveraging hindsight labeling to infer goals. As a result, the approach in [1] is not applicable to the data regimes considered in our work.
>
> To teach the LLM [1] uses DPO (with both positive and negative trajectories in a contrastive way) while we only teach the positive trajectories via SFT. This alternative approach is interesting and could be used in TEDUO step 3. We are happy to cite this work in the related section of the updated version of our manuscript.
>
> ---
>
> Thank you once again for your feedback. We’re thrilled by your excitement about TEDUO’s potential and hope these revisions solidify its contribution. Thank you for your support!

---

> > ### Comment · Reviewer_vbjR · 2025-04-01
> >
> > Dear Authors,
> >
> > Thanks a lot for your thoughtful response! This mostly satisfies my concerns. I have just one followup question:
> >
> > > In the second part, they shift to online interaction, generating trajectories from their trained models, whereas TEDUO remains strictly offline throughout.
> >
> > Could you explain this part? I did not think [1] uses the trained model to generate more data to train it further, but I am curious if I do not understand this difference between your work and this paper correctly.
> >
> > Thanks!

---

> > > ### Author Response · Authors · 2025-04-02
> > >
> > > Dear reviewer,
> > >
> > > We are glad that our previous response addressed your last concerns.
> > > Regarding the use of generated trajectories during training in [1], our understanding is based on Section 3.4, *"Scalable online curriculum learning."* In this section, the authors describe evaluating their policy through online interaction in order to estimate task difficulty: *"we then uniformly sample one task from the chosen group to evaluate the model performance with C rollouts."* While [1] does aim to reduce the number of rollouts required, acknowledging the associated costs, the training process still relies on an online setting.
> > >
> > > We hope this clarifies the difference, and we are happy to discuss further if needed.

---

### Official Review · Reviewer_azTQ · 2025-03-18

**Overall Recommendation:** 1

**Summary:**

This paper introduces TEDUO, a training framework that aims to enhance language-conditioned policy learning in autonomous agents while reducing the reliance on extensive data. The framework is structured around three key stages, each leveraging the capabilities of large language models (LLMs).

First, data enhancement is performed by using LLMs to abstract states and apply hindsight labeling to an unlabeled dataset of state-action transitions, resulting in the creation of labeled datasets. Next, policy learning is conducted by utilizing offline reinforcement learning (RL) algorithms to develop optimal policies tailored to a finite set of training goals. Finally, the framework emphasizes generalization, where a base LLM is fine-tuned to encode knowledge of environment dynamics and optimal actions, enabling it to adapt to unseen states and interpret novel language commands effectively.

**Claims And Evidence:**

The claims are generally well-supported.

**Essential References Not Discussed:**

N/A

**Experimental Designs Or Analyses:**

1. More experiments on complicated tasks (with diverse goals, and larger action space) to prove the applicability of this method is necessary. Please see Methods for details.

2. Comparison to more simplified RL methods. e.g. Deepseek R1 like methods that directly use a task success verifier as reward to train the policy model.

**Methods And Evaluation Criteria:**

1. The pipeline for data enhancement is reliant on the symbolic nature of the tasks. For example, BabyAI is a easy to define symbolic task where changing the shape or color of objects and doors would result in new tasks. Also Webshop tasks are primarily focused on browsing objects, with actions mainly limited to search [A] and buy [A]. However, this would be more difficult to enhance data for more complex environments, e.g. Webarena, Osworld.

2. Also, the method lacks scalability due to reliance on memory-intensive tabular Q-learning. It would suffer if the task is very complicated and requires many steps to finish.

3. Rewriting trajectories as data enhancement is very limited due to the extent of change you could perform when rewriting trajectories. For example, no additional exploration and trial and error are added.

**Other Comments Or Suggestions:**

N/A

**Other Strengths And Weaknesses:**

N/A

**Questions For Authors:**

1. Could you elaborate on how you could create abstract states and templates for a quite complicated task, e.g. webarena? Note that the website for browsing could be written as abstract state, but the next state after clicking on the website could be very different.

2. Direct RL training has been shown more effective in generalization than trajectory SFT in recent work [1]. Why is it necessary to use abstract template generated trajectories for SFT rather than directly perform RL training.


[1] SFT Memorizes, RL Generalizes: A Comparative Study of Foundation Model Post-training.

**Relation To Broader Scientific Literature:**

The method proposed by the paper is novel, yet it's conclusion is not solid due to lack of comparison on more complicated tasks and up-to-date baseline models. Also the improvement in generalization ability is limited to generalization to similar environments and lacks discussion of generalization to more challenging tasks (e.g. pick up red ball -> pick up green ball, v.s. pick up red ball -> boss level in BabyAI).

**Theoretical Claims:**

No theoretical claims are involved.

---

> ### Author Rebuttal · Authors · 2025-04-01
>
> Thank you for your review and constructive feedback. We appreciate your engagement with our work and have carefully addressed your concerns below:
>
> ---
>
> ### P1 Task Complexity and Scalability
>
> See P2 in the answer to reviewer `zuF6`, including **new results on generalization from simple to complex tasks**.
>
> ### P2 Scalability of Q-Learning
>
> Tabular Q-learning was used in our main experiments as it suits the BabyAI environment. However, TEDUO’s pipeline is agnostic to the choice of the learning algorithm in step 2. For larger state spaces, it can incorporate more scalable offline RL methods like DQNs, which our ablation study (Appendix B.4) shows perform comparably on BabyAI. In WebShop, we instead used filtered behavioral cloning, as Q-learning would be ineffective due to the limited number of trajectories in comparison to the dimensionality of the state space. WebShop was intentionally chosen to demonstrate TEDUO’s flexibility, reinforcing that it is agnostic to the policy-learning algorithm.
>
> ### P3 Trajectory Rewriting and Offline RL
>
> While trajectory rewriting indeed cannot introduce new exploration, development of flexible offline RL methods is critical for real-world settings where exploration is unsafe or impractical (e.g., robotics, healthcare). TEDUO’s focus on *offline* training aligns with such requirements. That said, we find the extensions of similar hybrid approaches employing LLMs as RL agents in online settings a highly promising area of research, which is beyond the scope of this work.
>
> ### P4 Additional RL baselines and comparison to DeepSeek R1
>
> We agree that comparisons to state-of-the-art baselines are essential. However, as TEDUO focuses on *offline* language-conditioned RL, to the best of our knowledge, there are no directly comparable prior works beyond the ones already included in our benchmarking. We are open to suggestions regarding additional baselines suited for the offline language-conditioned RL setting. Regarding generalization to more complex tasks, we have now extended our evaluation with a new experiment demonstrating our method’s success (see P1). Please note, testing RL agents on more complex tasks than the ones seen at training time is non-conventional and we find the presented result very exciting.
>
> **Comparison to DeepSeek R1.** The reviewer rightly notes the success of recent works employing GRPO for *online* policy improvement in the context of LLM training. However, such methods require access to real-time experimentation to enable policy rollouts. The focus of this paper is on learning from passively collected observational datasets. We find employing ideas similar to the ones observed in Deepseek R1 a very promising direction for follow-up works.
>
> - **New result:** We included a comparison of TEDUO to the baseline of prompting Deepseek R1, see P1 in the answer to reviewer `zuF6`.
>
> ### P5 Question: State abstraction
>
> We agree with the reviewer’s suggestion to clarify the role and implementation of state abstractions. State abstraction serves two purposes: (1) transforming the states from their initial modality into text and (2) filtering all irrelevant information to achieve the goal. For a web agent like Webshop or WebArena, the first purpose is achieved by transforming the HTML code of the webpage into a curated text report, where possible actions (button/search bar, etc.) are identified. The second purpose is achieved by removing irrelevant information such as advertisements or clearly irrelevant buttons etc. In our case for Webshop, the first part is natively supported by the environment, and the second part is not needed due to the noiseless nature of the provided state.
>
> ### P6 Question: Direct (online) RL training
>
> TEDUO prioritizes *offline* RL because many real-world applications (e.g., healthcare, education) prohibit online exploration. While online RL excels in simulated settings, our framework addresses practical constraints. The cited work ([1]) focuses on online RL, which is orthogonal to our setting.
>
> [1] SFT Memorizes, RL Generalizes: A Comparative Study of Foundation Model Post-training
>
> ---
>
> Thank you for suggesting the areas for improvement, which has helped us refine and better articulate TEDUO’s contributions. Your points inspired us to:
>
> - **Push generalization further** with new results showing the promise in TEDUO’s ability to train LLM agents to generalize to new, more complex tasks.
> - **Clarify scalability** by emphasizing TEDUO’s compatibility with more scalable offline RL algorithms, like DQNs.
> - **Highlight the focus** on real-world data-constrained applications and discuss exciting areas for **future work,** extending LLM+RL hybrids to online settings.
>
> We hope you’ll find our revisions compelling and reconsider your score. Thank you for helping us in making a meaningful contribution to the field.

---

> > ### Comment · Reviewer_azTQ · 2025-04-04
> >
> > Thank you for your detailed explanations. I’ve reviewed your added experiments on a more complicated BabyAI task. However, my main concern remains whether offline data collection on simpler tasks has the potential to generalize effectively to more complicated scenarios. While data diversity may increase, the complexity of the collected trajectories might not vary significantly, which could lead to limited generalization, which has been addressed in previous work [1]. Compositional generalization—generalizing from simpler tasks during training to more complex tasks—continues to be a key challenge, particularly in symbolic tasks.
> >
> > Regarding the focus on offline RL, your explanation partially addresses my concerns. It would be helpful to explicitly highlight the focus and discuss the scope of TEDUO in this context, especially in comparison to existing works in offline RL. However, given the limitations of this work’s scope to symbolic tasks, and considering that there are already more general online and offline methods (e.g., DPO), I remain inclined towards rejection.
> >
> > [1] Yuan, Lifan, et al. "Advancing llm reasoning generalists with preference trees." arXiv preprint arXiv:2404.02078 (2024).

---

> > > ### Author Response · Authors · 2025-04-07
> > >
> > > Thank you for your thoughtful comments and for engaging deeply with our work. We appreciate that our previous response helped address some of your concerns, and we would like to further clarify and contextualize the remaining points you raised.
> > >
> > > **On the generalization capabilities from offline data:**
> > > We agree with the reviewer that the ability to generalize to more complex tasks is fundamentally constrained by the information content of the collected trajectories. However, we view this as a challenge inherent to offline RL rather than a weakness of our particular method. TEDUO aims to address this challenge by leveraging external knowledge from LLMs, both to label trajectories and to abstract the state space.
> > >
> > > **On the motivation for offline RL and comparison to online methods:**
> > > The use of an offline RL framework is not merely a design choice, but a necessity dictated by many real-world applications, such as finance, education, autonomous driving, and healthcare, where online data collection can be costly, unsafe, or legally constrained. Therefore, while we appreciate the comparison to online RL methods such as [1], these are not interchangeable with offline methods in such settings. Offline and online RL serve fundamentally different use cases, and direct comparison can be misleading.
> > >
> > > **On the role of DPO and relevance to TEDUO:**
> > > We understand the reviewer's suggestion that DPO may be seen as a more general alternative. While DPO is indeed used in RLHF settings to fine-tune large language models from preference data when an absolute reward function is unavailable, we would like to clarify its relevance in our context. Specifically, DPO could only be applied to Step 3 of our pipeline, where the LLM is taught to reproduce optimal policies using SFT. However, in our case, an explicit reward function is available, making the application of DPO less natural or necessary.
> > >
> > > Moreover, the work cited by the reviewer ([1]) explicitly shows that in online RL settings, DPO performs strictly worse than SFT for distilling policies. This suggests that, even within its applicable scope, DPO may not offer empirical advantages in our setting.
> > >
> > > More importantly, the core contribution of TEDUO is orthogonal to this comparison: our work investigates the generalization ability of the full training pipeline, from unlabelled data to policy generation applicable across diverse tasks. This broader focus--particularly the transformation of unlabelled data into trainable signals--is not addressed by DPO or similar methods.
> > > We hope this clarification helps position our contribution more precisely within the broader landscape of RL.
> > >
> > > **On compositional generalization and task complexity:**
> > > The reviewer raised an important point about compositional generalization. While we provide preliminary evidence that TEDUO can generalize from simpler to more complex tasks in section 5.3 and with the new experiments, we emphasize that the generalization to out-of-distribution goals is beyond the typical expectations of goal-conditioned RL, where tasks at test time are usually drawn from the same distribution as training tasks [2], including in terms of difficulty.
> > >
> > > **On clarifying our offline RL focus:**
> > > Thank you for suggesting we clarify our focus. We will revise the introduction to more clearly motivate the choice of the offline RL setting. Additionally, detailed discussions of related work in goal-conditioned offline RL can be found in Sections A.1 and A.2 of the appendix. We would be happy to incorporate any other relevant references you believe are essential.
> > >
> > > **On the symbolic environment constraint:**
> > > We acknowledge, as noted in our limitations section, that the proposed instantiation of TEDUO is restricted to environments that can be represented symbolically. However, we believe this limitation is mitigated by the broad expressiveness of natural language.
> > > - Many environments studied in autonomous agent research inherently offer textual state representations, such as web or computer environments (as proposed by the reviewer) [3,4], video games [5], and multi-turn dialogue systems [6].
> > > - For most reinforcement learning environments, which often have tabular representations, these can be converted into key-value pairs. Values can be discretized as needed, which is a common practice in RL.
> > > - In the case of pixel-based RL, TEDUO could leverage VLMs instead of LLMs for both the data enhancement and fine-tuning stages. Alternatively, converting pixel-based environments into textual representations is an active area of research [7].
> > >
> > > Given the above points, we believe that focusing our attention on environments representable in a text format is not a major limitation of the proposed approach.
> > >
> > > [1] Advancing llm reasoning generalists with preference trees, 2024
> > >
> > > [2] Goal-Conditioned Reinforcement Learning: Problems and Solutions, 2022
> > >
> > > [3] OSWorld, 2024
> > >
> > > [4] WebArena, 2024
> > >
> > > [5] NetHack, 2020
> > >
> > > [6] LMRL Gym, 2023
> > >
> > > [7] ALFWorld, 2021

---

### Decision · Program_Chairs · 2025-05-01

**Decision:**

Accept (spotlight poster)

**Comment:**

This paper introduces TEDUO, a training framework designed to enhance language-conditioned policy learning in autonomous agents while reducing reliance on large-scale labeled data. The framework proceeds in three stages, each leveraging the capabilities of large language models (LLMs). First, LLMs are used to abstract states and apply hindsight labeling to an unlabeled dataset of state-action transitions, generating labeled data. Second, offline reinforcement learning (RL) algorithms are employed to learn optimal policies for a finite set of training goals. Finally, the framework focuses on generalization by fine-tuning a base LLM to encode environment dynamics and optimal actions, enabling adaptation to novel language instructions and unseen states.

As noted by Reviewer vbjR and others, there are several existing approaches that use LLMs to enhance the generalization of RL policies. Beyond those cited in the paper, related methods from the Decision Transformer family—such as Pre-trained Language Models Improve the Few-shot Prompt Ability of Decision Transformer; Meta-DT: Offline Meta-RL as Conditional Sequence Modeling with World Model Disentanglement—also leverage pre-trained transformers for in-context learning across tasks. Including a discussion of such work in the related literature would help further contextualize the proposed method.

Despite the breadth of existing literature in this area, the reviewers found the approach in this paper—using LLMs to transform classical RL problems into a text-based representation and construct solvable MDPs—to be novel and refreshing. The paper is clearly written and well-presented, with the main claims supported by thorough ablation studies.

A minor limitation is that the experiments are conducted in relatively simple environments with modest task variation. Nonetheless, the results are solid and convincing, and the method shows potential for broader impact. Overall, I agree with the reviewers that this is a strong submission. I recommend a strong acceptance of this work.